# Division of Labour and Parental Mental Health and Relationship Well-Being during COVID-19 Pandemic-Mandated Homeschooling

**DOI:** 10.3390/ijerph192417021

**Published:** 2022-12-18

**Authors:** Mariam M. Elgendi, Sherry H. Stewart, Danika I. DesRoches, Penny Corkum, Raquel Nogueira-Arjona, S. Hélène Deacon

**Affiliations:** 1Department of Psychology and Neuroscience, Dalhousie University, Halifax, NS B3H 4R2, Canada; 2School of Psychology, University of Sussex, Falmer, Brighton BN1 9QH, UK

**Keywords:** division of labour, COVID-19, homeschooling, equity, specialisation, parents

## Abstract

While the COVID-19 pandemic has impacted the way parents partition tasks between one another, it is not clear how these division of labour arrangements affect well-being. Pre-pandemic research offers two hypotheses: economic theory argues optimal outcomes result from partners specialising in different tasks, whereas psychological theory argues for a more equitable division of labour. The question of which approach optimizes well-being is more pressing in recent times, with COVID-19 school closures leaving many couples with the burden of homeschooling. It is unknown whether specialisation or equity confer more benefits for mandated homeschoolers, relative to non-homeschoolers or voluntary homeschoolers. Couples (*n* = 962) with children in grades 1–5 completed measures of workload division and parental well-being. A linear mixed modelling in the total sample revealed that specialisation, but not equity, promoted increased parental emotional and relationship well-being. These relations were moderated by schooling status: voluntary homeschoolers’ well-being benefitted from specialisation, whereas mandated homeschoolers’ well-being did not benefit from either strategy; non-homeschoolers well-being benefitted from both strategies. Across the mixed-gender couples, mothers’ and fathers’ well-being both benefitted from specialisation; equity was only beneficial for mothers’ well-being. Overall, couples might be advised to adopt highly equitable and specialised arrangements to promote both parents’ well-being.

## 1. Introduction

The COVID-19 pandemic has caused disruptions to work and family life on a global scale. In March 2020, the World Health Organization (WHO) proclaimed the COVID-19 outbreak a global pandemic. Shortly thereafter, massive containment measures were implemented as part of efforts to curb the spread of the virus [1]. In North America, states and provinces issued stay-at-home orders, prohibited large gatherings, and closed businesses, workplaces, and schools. In compliance with these measures, many people were forced to limit their time outside home. For many parents, traditional care supports were suspended, leaving them to balance mandated homeschooling of their children with working from home during a time of already heightened uncertainty. Given these massive disturbances to work and family life, couples were forced to readjust their roles and responsibilities. Accordingly, this study intended to explore the impact of these new work and family life arrangements on the mental health and relationship well-being of parents during the COVID-19 pandemic, with a specific focus on how homeschooling due to the COVID-19 pandemic affected these relationships.

Many health experts have expressed that the psychological footprint of a large-scale pandemic such as COVID-19 is likely to far exceed the medical footprint of the disease [2]. Research has already shown the negative impacts of these containment measures on the mental health of couples [3], particularly those with children [4]. Women in particular were at an increased risk of developing mental health and substance use problems during the pandemic [5,6,7,8]. A recent meta-analysis uncovered that both being female and having children increased the likelihood of psychological distress in response to viral outbreaks [9].

One of the most important factors affecting couples’ mental health and relationship well-being is the division of labour [10,11]. Division of labour, defined as the partitioning and assignment of tasks to different individuals [12], is a particularly significant issue among couples with children, who must negotiate the division of both household and childcare tasks. During the pandemic, this division of labour also included the division of homeschooling responsibilities for many families. 

Homeschooling is a practice in which parents educate children at home rather than send them to school [13]. Traditionally, this has been a voluntary choice, but COVID-19 restrictions brought this to many families on a non-voluntary basis. Here, we use the term “Mandated homeschooling” to refer to the practice where parents are responsible for supervising and assisting their children to navigate school-related tasks and online learning at home due to COVID-19 school closures. Couples mandatorily homeschooling due to the COVID-19 pandemic are likely to experience more disruptions to their work–life arrangements compared to non-homeschoolers (i.e., parents whose children attend school in-person) or those voluntarily homeschooling their children (i.e., parents who chose to homeschool for reasons unrelated to the COVID-19 pandemic). Indeed, couples mandated to homeschool due to the COVID-19 pandemic have been shown to be less optimistic and more likely to engage in coping-related substance use than non-homeschooling couples [14]. Mandatory homeschooling was thrust upon parents with no notice and with varying degrees of support from education systems. Children have been forced to be at home for longer periods of time and require more support from their parents than usual. Consistent with this, there is a growing body of research highlighting that the COVID-19 pandemic has decreased paid work responsibilities and increased unpaid work responsibilities such as childcare and household management for both parents [15,16,17].

Experts, however, are conflicted on how the division of labour has been impacted by the COVID-19 pandemic. Some argue that women are disproportionately impacted by the increased unpaid work responsibilities such as homeschooling [17], while others suggest that the pandemic has provided an opportunity to reduce gender inequalities around paid and unpaid labour [16]. Further, it is not yet clear how these new arrangements affect parents’ mental health and relationship well-being, particularly among mandated homeschooling parents.

Pre-pandemic research offers two plausible theories as to what type of division of labour is most beneficial to parents’ mental health and relationship outcomes [11]. On the one hand, *economic theory* argues that partners specialising in performing particular roles and tasks improves mental health and promotes stronger and healthier relationships in comparison to sharing roles [18]. An example of specialisation is having one partner engage in household labour, while the other takes on paid labour. On the other hand, *psychological theory* argues that an equity of the division of labour is more beneficial than an inequitable division of labour in relationships, e.g., both partners put in the same time into unpaid and/or paid labour [19,20,21]. Equity and specialisation are often regarded as being mutually exclusive in the literature, particularly because of the increased focus on examining how couples divide one of the forms of labour, such as looking at household labour *or* childcare alone [22,23,24,25,26]. However, this is not true in practice given all of the responsibilities that couples are tasked with and must simultaneously balance. A couple can both specialise (in conventional or counter-conventional ways) and be equitable in their division of labour under some circumstances. For instance, this would be the case if each member of a couple spent the same number of hours working every week, but one exclusively participated in paid work while the other exclusively in childcare and/or household work. The scarce extant literature [27] examining the differential effects of equity and specialisation finds that these two strategies independently vary relative to one another, although they are negatively related (*r* = −0.26). Research also finds that couples adopt various combinations of these strategies (e.g., high levels of both equity and specialisation or high levels of equity and low levels of specialisation), although traditional arrangements are more common (e.g., high levels of specialisation and low levels of equity) [27]. Accordingly, the present study examines how specialisation and equity might each be separately predictive of emotional and relationship well-being by examining parents’ division of both paid and unpaid labour (childcare and household labour) simultaneously. In the sections below, we review each of these theories and examine how they fit in the broader context of the COVID-19 pandemic, particularly in the mandated homeschooling context.

### 1.1. Economic Theory—Specialisation

The economic principle of specialisation proposes that the division of labour in a relationship is most successful when couples specialise in different tasks [18]. Specialisation refers to when partners of a couple perform different activities; for example, a couple with high levels of specialisation might have one partner who contributes the majority of their time to paid labour, whilst the other mostly engages in unpaid labour. This theory was initially applied to explain the division of labour in the workplace and was later adapted to explain household dynamics [18,28]. Specialisation is based on the rationale that when partners devote their time to a specific type of task, they each develop unique and specific skills and motivations that maximise the output of the household [18]. Accordingly, it has been suggested that specialisation might have positive impacts on relationship functioning and couples’ mental health because it is more efficient than equity. For instance, in couples who specialise, only one member will have to spend time and mental effort attending to work demands, or must balance work and family commitments, e.g., devoting time to their child’s needs [29,30,31]. Theoretically, specialisation also posits that one partner may have a comparative advantage in certain types of tasks as a result of societal circumstances (e.g., gender-specific wage discrimination in the workplace, favourable tax policies for dependent spouses, women’s rising economic autonomy) and/or intrinsic biological differences (e.g., hormones and physical body structure) [18]. Moreover, less specialised couples are likely to have to bargain and coordinate more with their partner and are also likely to monitor each other’s efforts to ensure that each is equally contributing to the relationship [18,28]. Research shows that dual-income couples report more relationship conflict, distress, and time pressure than single-income couples [29,30], suggesting that having one partner who specialises in paid labour might confer more benefits for the couple. Further, Kalmijn et al. found that high levels of specialisation compared to lower levels of specialisation (i.e., more role-sharing arrangements) were protective against dissolution of marriage [32].

Although the economic principle of specialisation is theorised to be effective in improving relationship satisfaction regardless of which partner contributes to paid vs. unpaid labour, research has found more traditional arrangements of specialisation to be more beneficial, i.e., when men are responsible for all or the majority of paid labour [32]. Further, research consistently finds that higher levels of specialisation (i.e., less role-sharing) disproportionately benefits women’s well-being [31,33]. Although both parents in dual-income families reportedly spend an equal amount of time thinking about work and family responsibilities, women are more distressed by these thoughts [31]. Further, studies on dual-income couples show that mothers, as compared to fathers, report more instances of work interfering with family responsibilities and family interfering with work commitments, particularly when children younger than 12 years old are residing at home [33,34,35]. Increased levels of work–family interference have been associated with increased levels of physical (e.g., more health-related problems [36]), emotional (e.g., increased levels of depression [37]), and relational problems (e.g., increased levels of relationship conflict [38,39]). Further, the participation of women in the labour force and their income levels have increased more so than for men, with this shift being associated with an increased risk of divorce and separation [32,40,41,42]. A couple is less specialised when both partners spend more hours engaging in paid labour and share the burden of contributing to household income. Accordingly, the negative impacts of women’s participation in the workforce on relationship well-being has been theorised to be due to the adverse effects of reduced specialisation on couples’ well-being [43,44]. Collectively, this highlights important gender differences associated with adopting more specialised roles within a relationship.

In the context of the COVID-19 pandemic, some experts have argued that the disruptions of the pandemic are more likely to cause parents to adopt traditional gender roles, with mothers exclusively taking on the responsibility of increased housework and childcare and fathers assuming paid work responsibilities [15]. Recent parent testimonials in *Nature* [45] and *The New York Times* [46] support this conjecture. Indeed, reports from couples with young children during the first wave of the pandemic showed that the burden of increased household and childcare work had largely fallen to mothers [47]. On the paid work front, there is evidence that women were more likely than men to have reduced their working hours or to have lost their jobs due to the COVID-19 pandemic [48], leaving more time to spend at home caring for their families and homeschooling children [49].

However, it is not yet clear how specialisation among couples during the COVID-19 pandemic impacts mental health or relationship well-being. On the one hand, spending more time with children at home might have protective effects on parents’ mental health. Indeed, some pre-pandemic studies show that having children at home is associated with higher levels of overall well-being and lower levels of depressive symptoms [50,51,52,53,54,55,56,57]. On the other hand, the COVID-19 pandemic has increased unpaid and decreased paid labour tasks in the home [17]. Therefore, specialising in tasks during a pandemic may lead to one partner taking on a greater workload than the other, which can be detrimental to mental health and relationship well-being. A recent UN policy brief noted that the COVID-19 pandemic will increase both women’s home responsibilities [58] and conflict in the home, largely directed at women [59]. If women are expected to specialise in childcare and household work, including taking on the additional responsibility of mandatory homeschooling, with little to no support from the educational system and in the absence of their typical supports (e.g., shared time with families and access to libraries, parks, and gyms), this will likely have detrimental effects on their mental health [9].

### 1.2. Psychological Theory—Equity

The equity hypothesis predicts that an *equitable* division of labour will promote parental mental health and relationship well-being [19,20,21]. This hypothesis is built on the premise that a relationship is more agreeable when the division of labour is perceived to be just and fair. Indeed, research indicates that a person’s subjective interpretation of the distribution of work in a relationship has a greater impact on relationship well-being than the actual objective division of labour [60,61]. The equity hypothesis is derived from evaluations of the amount of work, time, and effort that both partners contribute into a relationship and the relative rewards that they gain. Accordingly, equity occurs in a relationship if the efforts one puts forth and the rewards one gains are equal to those of their partner. Following this rationale, an unequal division of labour, in which one person is doing drastically more than the other, can lead to negative impacts on mental and relational health [20]. A person who perceives themselves to be doing more than their partner is likely to develop negative emotions, such as frustration, anger, and sadness, which can quickly escalate into anxiety and depression [62]. Similarly, the under-involved partner can also develop anxiety and depression due to feelings of shame and guilt [62,63]. Furthermore, inequity can also lead to conflict in the relationship which, in turn, is likely to negatively impact mental health and relational satisfaction. Indeed, research has shown that inequity is associated with greater marital conflict and lower marital satisfaction [21,64,65], both of which contribute to poorer mental health [66]. Couples with more equitable divisions of labour also report higher levels of relationship well-being, including increased emotional intimacy and commitment [23,67,68]. Equity has also been associated with higher reported levels of perceived social support in romantic couples [68,69], which has been established to buffer the negative impact of stress on an individual’s mental health [70]. In contrast, inequity within a relationship is associated with an increased risk of marital disruption and dissolution, particularly if a woman believes she was contributing more to the relationship [71,72]. Reflecting this, longitudinal research suggests that inequity can have long-lasting negative effects on marital quality, with known effects persisting up to 20 years [73].

While the equity hypothesis predicts benefits for both members of a couple, research has underscored the gendered effects of equitable labour divisions at home. Specifically, women tend to have stronger beliefs than men that labour should be divided equally in a relationship and attribute more importance to these arrangements in a relationship [74]. Intuitively, these gender differences in beliefs around equity might arise from women’s continued experiences with inequity. Furthermore, while an inequitable division of labour negatively impacts both men and women, women’s experiences of inequity tend to be more intense and/or have more severe consequences on their psychological distress, role strain, and physical health [75,76,77,78,79,80]. Women are also more emotionally responsive to inequity than men, particularly when the division of labour benefits them more, and report experiencing more days of distress and negative emotions including fear and anger related to their experiences of inequity [81]. Collectively, this suggests that inequity places women at a greater disadvantage than men.

In the context of the COVID-19 pandemic, some experts predict that the imposed containment measures might lead to a more equitable division of labour [16,17,82]. Over the past several decades, research has shown a decreasing gender gap in couples’ division of labour [83]. This is reflected in men’s increased participation in household and childcare tasks and women’s increased involvement in paid labour. Despite this trend, pre-pandemic research suggests that women today still contribute more of their time and effort into household and childcare tasks compared to men, regardless of their employment status [84,85,86]. The COVID-19 crisis led many employees to adopt more flexible work arrangements, which might have exposed many fathers specifically to the increased burden of childcare responsibilities and household demands, which women may be more accustomed to balancing. This exposure led many fathers to become more involved with household and child-rearing tasks during the pandemic than before [16,17,82], which serves as evidence of a more equal sharing of domestic labour. This shift makes sense given that pre-pandemic research highlights that working from home is associated with a more equal sharing of household labour by men [87]. Moreover, COVID-19 containment measures eliminated many unsupportive workplace cultures and policies, e.g., inflexible scheduling or lack of paid family leave, which are not conducive to an equitable division of labour [88]. Indeed, studies on fathers taking parental leave suggests that sudden exposures to childcare and household management can have long-lasting effects on men’s participation in unpaid care work [89,90,91]. Intuitively, men sharing domestic labour more equally with their women partners during the pandemic might reduce the pressure put on women to take on additional domestic/childcare responsibilities and thus promote better mental health for both partners [92]. Accordingly, a more equitable division of labour during the COVID-19 pandemic might have protective impacts on parental well-being.

### 1.3. Homeschooling and Division of Labour

The impact of COVID-19-mandated homeschooling on the division of labour and on parent mental health has yet to be explored. Prior to the pandemic, most voluntary homeschooling parents (approximately 80%) reported that mothers were more likely to take on homeschooling their child, while fathers assumed more paid labour responsibilities [93,94,95,96]. Reflecting on the voluntary homeschooling literature [97], mothers who voluntarily homeschool their children frequently report experiencing intense role strain and emotional burnout [98,99,100]. In the context of the pandemic, the effects of such homeschooling arrangements might be exacerbated in couples mandated to homeschool due to COVID-19-associated school closures. Specifically, women who are mandated to homeschool their children may be at greater risk for psychological distress because they must balance these additional homeschooling responsibilities with their massively increased household workloads and reduced time and emotional capacity. This suggests that an equitable rather than a more specialised division of labour might be more beneficial for couples mandated to be responsible for educating their children at home. Indeed, preliminary research into the impact of mandated homeschooling on mothers vs. fathers seems to hint at a protective effect of fathers taking a greater share of the homeschooling responsibility on mothers’ alcohol use [92]. However, research has yet to compare voluntary and mandated homeschoolers, which are likely to have different ways of dividing labour and/or responding to labour arrangements. Accordingly, it is also important to distinguish between those mandated to homeschool and those who voluntarily take up this responsibility. Specifically, parents voluntarily homeschooling prior to the pandemic may already have household arrangements in place [101] that would serve to help them educate their children at home during the COVID-19 pandemic. However, the pandemic is also likely to have negatively impacted voluntary homeschooling parents, particularly because of decreased access to the supports on which voluntary homeschooling communities may normally rely upon, such as libraries, public gym facilities, and other like-minded homeschooling families [102,103]. However, the rapid onset of the COVID-19 pandemic and consequent unexpected school closures might put mandated homeschoolers, particularly mothers, at an increased risk for mental health and relationship problems. Accordingly, mandated homeschoolers might benefit more from protective factors such as an equitable division of labour relative to voluntary homeschoolers. In summary, while both equity and specialisation seem to confer specific advantages for parents in general (i.e., non-homeschooling couples), whether equity is more beneficial to homeschooling parents, particularly to women and those mandated to homeschool due to COVID-19, has yet to be determined.

### 1.4. The Current Study

The primary goal of this study was to examine how equity and specialisation *separately* predict well-being in parents during the COVID-19 pandemic and how this differs in mandated homeschooling couples relative to non-homeschooling and voluntary homeschooling couples. We further aimed to explore whether equity and specialisation predicted different outcomes for men and women among mixed-gender couples, given the differential impact of the division of labour on mothers and fathers in pre-pandemic research [72,77,79]. We intended to examine the impacts on both emotional (i.e., anxiety, depression, and stress) and relationship well-being (relationship conflict and satisfaction) to accurately capture effects on both individual- and couple-level stability, which are most likely to be impacted by the division of labour in the home [10,11]. Lastly, we also conducted supplementary analyses to explore potential three-way interactions between gender, homeschooling status, and specialisation/equity.

Consistent with predictions emerging from the equity and specialisation hypotheses, we expected: (**H1**) a greater specialisation of tasks between parents to predict better emotional and relationship well-being; (**H2**) a more equitable division of labour in parents to also predict better emotional and relationship well-being. Specifically, we expected more specialisation and equity to predict lower levels of depression, anxiety, stress, and relationship conflict, as well as higher levels of relationship satisfaction in parents, though this effect might be stronger for equity vs. specialisation [27]. Given that previous research suggests that the division of labour has more negative impacts for mothers than for fathers [72,75], we also expected mothers to benefit more from (**H3**) more specialisation and (**H4**) equity than fathers. Given the uncertainty and stress around homeschooling during the COVID-19 pandemic, we also expected (**H5**) parents of in-person learners (i.e., non-homeschooling parents) to benefit from both equity and specialisation, and we expected both mandated and voluntary homeschooling parents to benefit more from an equitable division of labour than from specialisation, with the former being more beneficial to a greater extent than the latter. 

## 2. Materials and Methods

### 2.1. Participants and Procedures

A total of 962 adult (19+ years old) Canadian (*n* = 806) and American (*n* = 156) romantic couples were recruited in March–May 2021 (In-person learners = 386 couples; Mandated homeschooling due to the COVID-19 pandemic = 332 couples; Voluntary homeschooling for reasons unrelated to the COVID-19 pandemic = 244 couples). The demographics information and relationship characteristics for the study sample are presented in Table 1.

Initially, 764 couples were recruited through Qualtrics Survey Panels, 332 couples of which were homeschooling due to COVID-19, 46 of which were homeschooling for reasons unrelated to COVID-19, and 386 of which were not homeschooling. Participants were first screened for eligibility and then asked to provide informed consent. Eligible respondents were those who had at least one school-aged child (Grade 1 to 5) living with them, and who were involved in a romantic relationship for at least 3 months with a partner who was also willing and available to participate. Both partners had to be: at least 19 years old, living in Canada or the US, and cohabiting together during the one month period of 15 January to 15 February 2021. Following this, respondents were asked to complete a series of measures, which included two filter questions to ensure that the respondent was paying attention (e.g., “Please select disagree”). The panelist completed the survey first before handing their device to their partner to complete the same set of measures (the partner had to complete the survey within 48 h of the panelist). Qualtrics also performed a speeder check (i.e., checking to identify responses that were more than two standard deviations from the mean duration) to ensure that adequate time was spent completing the survey. Couples who were ineligible, failed the filter questions or speeder check, or failed to provide consent, were screened out. Finally, after completing the survey, couples were compensated via Qualtrics according to their guidelines.

To supplement the recruitment by Qualtrics, 198 couples homeschooling for reasons unrelated to COVID-19 were recruited (during March–May 2021) from homeschooling associations and social media homeschooling groups. This was needed because our recruitment strategies through Qualtrics did not adequately capture this subset of homeschoolers (This was noted early in recruitment. Accordingly, this second method of data collection was instituted early in the data collection process and substantially overlapped with Qualtrics Survey Panels’ recruitment). These participants followed the same procedures outlined above (i.e., Qualtrics software was used to collect and link de-identified data for these parents across couples, and the same data quality screening measures such as speeder checks were utilised). The only exception was that participants were directed to a different location at the end of the survey, where they provided their email addresses so that they could be compensated ($10 Amazon gift cards for each member of the couple) by a co-investigator or trained research assistant. The participant email addresses were not stored with the data to help protect confidentiality. Lastly, this study received approval from Dalhousie University’s Social Sciences Research Ethics Board.

### 2.2. Sample Size Justification

Participants were part of an existing longitudinal study on the factors that support and impede family well-being during the COVID-19 pandemic. A post hoc power analysis also revealed that our sample was adequately powered (power of approximately 1) to detect a small effect size (*β* = 0.2) of equity/specialisation on parent well-being (**H1**–**H2**), assuming an alpha of 0.01. Our estimate of effect size was based on the results of extant literature on the effects of the division of labour on emotional and relationship well-being [11,27]. To assess if we had a sufficient sample size to detect moderation by gender or homeschooling status (**H3**–**H5**), we conducted additional post hoc power analyses (one assuming a sample size of 831 mixed-gender couples and one assuming a sample size of 962 couples). These analyses yielded similar findings (i.e., power of approximately 1). Given the complexity of estimating power for multilevel models, these calculations were based upon power analyses with actor–partner interdependence models (APIMPowerR) [104].

### 2.3. Measures

Both members of the couple completed the same set of measures in March–May 2021 using the same one month time frame as a reference (i.e., 15 January–15 February 2021. This timeframe was chosen as the only time during the pandemic when all three forms of schooling were taking place in both Canada and the US (Qualtrics targeted specific regions in both Canada and the US to attempt to ensure that our sample had adequate numbers of all three forms of schooling).

#### 2.3.1. Demographics

The demographics questionnaire asked about the respondents’ homeschooling status, gender, age, race, number and age of children, relationship status, family income, highest level of education, employment, ethnicity, and length of relationship. These were used in the sample description and some were used as covariates (i.e., age, number of children, relationship length, family income, and highest level of education), given previous research linking these variables with levels of emotional and relationship problems [105,106,107,108,109]. Further, these variables were significantly different across our three schooling conditions (see Table 1; *p* < 0.05) and were significantly and/or marginally associated with emotional and/or relationship well-being in bivariate correlational analyses in the current study.

#### 2.3.2. Workload Division Measure

Division of labour was assessed with questions on paid and unpaid service from the General Social Use survey (GSU) [110], to which we added questions on homeschooling labour (e.g., “Last week, how many hours did you spend looking after one or more children in your household?”). In line with past research [11,27], equity and specialisation were calculated for each couple as follows ($P_{a}$ [paid hours of partner *a*], $P_{b}$ [paid hours of partner *b*], $U_{a}$ [unpaid hours, i.e., household and childcare hours, of partner *a*], and $U_{b}$ [unpaid hours, i.e., household and childcare hours, of partner *b*]. Total number of hours is defined as $T_{ab}=P_{b}+U_{b}+P_{a}+U_{a}$):(1)$$\mathrm{Equity}=1-\frac{\left|\left(P_{a}+U_{a}\right)-\left(P_{b}+U_{b}\right)\right|}{T_{ab}}$$
(2)$$\mathrm{Specialisation}=\left|\left(\frac{P_{a}}{P_{a}+U_{a}}\right)-\left(\frac{P_{b}}{P_{b}+U_{b}}\right)\right|$$

Equity was defined as the absolute total difference in the proportion of time spent by both partners on paid and unpaid labour; to ensure that higher values represent more equity, scores were subtracted from 1. In contrast, specialisation was conceptualised as the absolute total difference in the proportion of hours that each partner spends on paid work relative to the total amount of work [27]. Scores could range from 0 to 1 for equity (0 = no equity; 1= full equity) and specialisation (0 = full role-sharing; 1 = full role specialisation). For homeschooling couples, hours spent homeschooling was included as part of childcare hours, which was part of unpaid labour.

#### 2.3.3. Emotional Well-Being

The 7-item Generalized Anxiety Disorder (GAD-7) scale [111] was used to assess generalised anxiety symptoms (e.g., “Feeling nervous, anxious, or on edge”) and the 9-item Patient Health Questionnaire (PHQ-9) [112] was used to assess depressive symptoms (e.g., “Feeling down, depressed, or hopeless”). Items on both measures were answered on a 4-point rating scale, ranging from “Not at all” (scored as 0) to “Nearly Every Day” (scored as 3), for the period of interest of 15 January–15 February 2021. For each measure, items were summed to create a total score. These anxiety [111,113] and depression [112] measures have good internal consistency (α*-values* > 0.89) and test-retest reliability, as well as good procedural and criterion-related validity across numerous populations, e.g., patients in primary care, patients with medical comorbidities, and cognitively impaired patients [114]. The PHQ-9 and GAD-7 tools are widely used as brief severity measures for depressive and anxious symptoms, respectively [114]. Both scales showed excellent internal consistency (α = 0.93 [GAD-7]; α = 0.91 [PHQ-9]) in the present sample. We also assessed perceived stress using the 4-item Perceived Stress Scale (PSS-4) [115]. This scale asked parents to rate how often they experienced stressful thoughts and feelings during the month of interest from 15 January–15 February 2021 (e.g., “In the last month, how often have you felt that you were unable to control the important things in your life?”). Items were answered on a 5-point rating scale ranging from “Never” (Scored as 0) to “Very Often” (scored as 4). The PSS-4 tool has been demonstrated to be an internally consistent and valid measure of stress in a sample of pregnant women and across a sample of adults in the UK (α = 0.77) [116,117]. Though the PSS-4 showed a lower internal consistency (α = 0.58) in the present sample, this is very close to the acceptable cut-off of 0.60 for short scales of 10 items or less [118].

#### 2.3.4. Relationship Well-Being

Relationship well-being was assessed using two scales, which tapped relationship conflict and relationship satisfaction, respectively. Relationship conflict was assessed with the 7-item Partner-Specific Rejecting Behaviors Scale (PSRBS) [119], completed for conflict with partner (e.g., “I was angry or irritated with my partner”). Items were answered on a 9-point scale ranging from “Strongly disagree” (scored as 1) to “Strongly agree” (scored as 9) for the period of interest of 15 January–15 February 2021. Items were summed to create a total conflict score for each member of the couple. This scale has been shown to possess excellent psychometric properties (i.e., internal consistency, α*-values* > 0.93; test-retest reliability, and validity) in prior studies using romantic dyads [120]. Relationship satisfaction was assessed using the 4-item dyadic Couples Satisfaction Index (CSI) [121]. Three items assess the level of agreement with the partner (e.g., “I had a warm and comfortable relationship with my partner”). These items were answered on a 6-point rating scale ranging from “Not at all” (scored as 0) to “Completely true” (scored as 5) for the one month period of interest. One item assesses the degree of happiness with the relationship, which was answered on a 7-point scale ranging from “Extremely unhappy” (scored as 0) to “Perfect” (Scored as 6). The CSI tool has been demonstrated to be reliable in romantic couples [122]; it highly correlates with other relationship satisfaction measures and discriminates between distressed and non-distressed relationships, establishing its validity [121]. Both scales showed excellent internal consistency (α = 0.95 [PSRBS]; α = 0.90 [CSI]) in the present sample.

### 2.4. Statistical Analyses

All analyses were carried out using R software version 4.1.2. The five outcome variables were initially screened and inspected for the approximate normality of residuals, linearity, constant variance, and extreme outliers. There were some violations of the normality and constant variance assumptions. Thus, we ran all the models using more robust estimation methods and computed heteroscedastic consistent standard errors [123,124]. Our results were unchanged.

In terms of testing our hypothesis, separate analyses were conducted for each predictor (i.e., equity and specialisation) and outcome variable (i.e., depression, anxiety, stress, relationship satisfaction, and relationship conflict). In all models, parent age, the number of children, relationship length, family income, and highest level of education were included as covariates, and all predictors were grand mean-centred. Due to the dyadic nature of the data, we used linear mixed models (LMMs) with indistinguishable dyads to test the impact of specialisation and equity on the emotional and relationship well-being of couples in the full sample (**H1** and **H2**). For each outcome variable, equity (or specialisation) and covariates (see above) were modelled as fixed effects while the participants were modelled as random effects nested within couples. An example of a model specification looking at the impact of equity on depression is as follows: Depression = β_00_ + β_01_ (Equity) + β_10_ (Covariates) + r_0_ + e. LMMs with gender-distinguishable dyads in the mixed-gender couples only were then used to assess gender differences on the impact of specialisation and equity (**H3** and **H4**). To assess if the impacts of equity and specialisation were different for homeschooling (voluntary and COVID-19-mandated homeschoolers) and non-homeschooling couples, we conducted all of the above analyses using homeschooling status as a moderator (**H5**). All moderators were effect coded and interactions were probed by separately estimating the simple slopes of equity and specialisation on the outcomes for different levels of the moderators (Gender: Male, Female; Homeschooling Status: In-person learners, Mandated homeschoolers due to COVID-19, Voluntary Homeschoolers). Model fit statistics (Bayes information criterion [BIC], Akaike information criterion [AIC], and R^2^) are presented for each model. Further, given our multiple outcome measures (*n* = 5), we used a more stringent alpha for determining statistical significance. Thus, all effects significant at *p* < 0.01 were considered statistically significant and discussed as so. However, as this was also a novel study (the first of its kind during the pandemic), we wanted to ensure that we also protected against type 1 error, and thus we note that all effects that are significant at traditional levels of significance *p* < 0.05 and discuss all effects between *p* < 0.05 and *p* = 0.01 as trends in the data.

## 3. Results

### 3.1. Descriptive Statistics and Bivariate Correlations

Descriptive statistics and bivariate correlations between all study variables are presented in Table 2. Overall, couples on average reported high levels of equity and low levels of specialisation. Further, based on population norms, parents in our sample endorsed mild symptoms of depression and anxiety but moderate to high symptoms of perceived stress [112,125,126,127]. Lastly, reports of both relationship conflict and satisfaction were higher (i.e., greater than 1 SD) than previous samples of romantic dyads [120,128,129].

With regard to bivariate correlations, correlations for the entire sample are presented below the diagonal, and the correlations for the mandated homeschooling sample are presented above the diagonal. In the whole sample, equity was only negatively correlated (marginally) with perceived stress, and specialisation was negatively correlated with all of our outcome variables (except relationship satisfaction). In the mandated homeschoolers sample, equity was marginally and negatively associated with depression and perceived stress, and specialisation was not correlated with any of our outcome variables. Equity and specialisation were significantly negatively intercorrelated in the whole sample, but the magnitude of this intercorrelation decreased in the mandated homeschooling sample.

### 3.2. Effect of Specialisation and Equity (**H1** and **H2**)

Consistent with the specialisation hypothesis (**H1**), a more specialised division of labour across parents was associated with significantly lower depression, anxiety, perceived stress (marginal), and relationship conflict (see Table 3).

Contrary to **H2**, a more equitable division of labour was not associated with any of our parental well-being outcomes (see Table 4).

### 3.3. Gender Differences in Specialisation and Equity (**H3** and **H4**)

Gender significantly moderated the effect of specialisation on all measures of parents’ emotional and relationship well-being (see Table 5); however, these effects were not as predicted. A simple slopes analysis was used to probe the interactions at both levels of the moderator: Male and Female.

Contrary to **H3**, more specialisation of tasks across parents was significantly associated with decreased depression (*β* = −3.82; *p* < 0.001; see Figure 1) and anxiety (*β* = −2.99; *p* < 0.001) in males and only marginally associated with decreased anxiety in females (*β* = −1.26; *p* = 0.04). Specialisation was also associated with decreased perceived stress (*β* = −1.12; *p* = 0.003) and relationship conflict (*β* = −5.16; *p* < 0.001) in males but not females (perceived stress [*β* = 0.02; *p* = 0.968]; relationship conflict [*β* = −1.35; *p* = 0.44]). Lastly, specialisation was not significantly associated with relationship satisfaction for either males (*β* = 0.38; *p* = 0.51) or females (*β* = −0.90; *p* = 0.12). However, the opposing effects of specialisation on males’ and females’ relationship satisfaction is likely driving the observed interaction.

Gender moderated the effect of equity on depression (only marginally), anxiety (only marginally), and relationship satisfaction, but not of equity on perceived stress or relationship conflict in parents in mixed-gender couples (see Table 6 and Figure 2).

Consistent with **H3**, simple slopes analysis indicated that in females, a more equitable division of labour was associated with decreased depression (*β* = −2.00; *p* = 0.02) and increased relationship satisfaction (*β* = 2.35; *p* < 0.001; see Figure 2) but was not associated with anxiety symptoms (*β* = −1.16; *p* = 0.13). There were no significant associations between being male and any of these outcomes (depression [*β* = −0.09; *p* = 0.92]; anxiety [*β* = 0.59; *p* = 0.44]; relationship satisfaction [*β* = 0.34; *p* = 0.63]). Despite the nonsignificant values, the opposing effects of equity on males and females anxiety is likely driving the observed interaction. Further, gender did not moderate the effect of equity on relationship conflict.

### 3.4. Differences by Homeschooling Status (**H5**)

With regard to the effects of specialisation on parents’ well-being, homeschooling status significantly moderated the effect of specialisation (Table 7) on parental depression, anxiety, and relationship satisfaction. Homeschooling status also marginally moderated the effect of specialisation on relationship conflict. Simple slopes analyses were used to probe the interactions at all levels of the moderator: In-person learners, Mandated Homeschoolers, and Voluntary Homeschoolers.

In contrast to **H5**, voluntary homeschoolers experienced significantly less depression (*β* = −6.80; *p* < 0.001), anxiety (*β* = −6.84; *p* < 0.001; see Figure 3), and relationship conflict (*β* = −12.36; *p* < 0.001) and reported significantly higher relationship satisfaction (*β* = 3.27; *p* < 0.001) with higher, as compared to lower, levels of specialisation across tasks. Similarly, but to a lesser degree, in-person learners also reported marginally less depression (*β* = −2.37; *p* = 0.01) and relationship conflict (*β* = −4.58; *p* = 0.04) with higher vs. lower specialisation. However, none of the associations between specialisation and measures of parents’ emotional and relationship well-being were significant for mandated homeschoolers (all *p*-values > 0.05).

Homeschooling status also significantly moderated the effect of equity on relationship satisfaction (Table 8), but not on other measures of parent well-being (all *p*-values > 0.05).

Partially consistent with **H5**, simple slopes analysis indicated that there was a marginally positive association between relationship satisfaction and equity for in-person learners (*β* = 2.09; *p* = 0.02; see Figure 4). In contrast to **H5,** the opposite trend was observed in voluntary homeschoolers; relationship satisfaction and equity were marginally negatively associated (*β* = −2.99; *p* = 0.04). Equity and relationship satisfaction were positively associated in mandated homeschoolers, but this was not significant (*β* = 1.73; *p* = 0.10).

### 3.5. Supplementary Analyses

Supplementary analyses were run to test potential three-way interactions between gender, homeschooling status, and specialisation/equity. There was a significant three-way interaction between specialisation, parent gender, and homeschooling status with regard to relationship conflict (Table 9; see Figure 5), but not any other outcomes. A simple slopes analysis was used to probe the interaction at all levels of the moderators: Homeschooling status (In-person learners, Mandated Homeschoolers and Voluntary Homeschoolers) and Gender (Male, Female). Men from voluntary homeschooling couples had a significant negative relationship between specialisation and relationship conflict (*β* = −21.34; *p* < 0.001). All other simple slopes were non-significant (all *p*-values > 0.05).

There was also a significant three-way interaction between equity, parent gender, and homeschooling status with regard to parent depression (only marginally) and relationship conflict (Table 10; see Figure 6 and Figure 7), but not any other outcomes. Simple slopes analyses indicated that female parents of in-person learners (*β* = −3.25; *p* = 0.01) had marginally significant negative associations between equity and depression (see Figure 6); this association was not significant in male and female mandated homeschoolers (*β* = −2.24; *p* = 0.13; *β* = −2.80, *p* = 0.06) as well as male parents of in-person learners (*β* = 0.34; *p* = 0.79). Equity and depression were not significantly associated in male or female voluntary homeschoolers (*β* = 0.01, *p* = 1.00; *β* = 0.63, *p* = 0.74). With regard to relationship conflict, male and female voluntary homeschoolers reported opposing effects of equity (*β* = 3.80, *p* = 0.42; *β* = −8.99, *p* = 0.06), although these relations were not significant. Further, there was a marginally significant negative association between equity and relationship conflict in male parents of in-person learners (*β* = −6.52, *p* = 0.04); this was similar but not significant in their female counterparts (*β* = −5.76, *p* = 0.07). Lastly, equity and relationship conflict were not significantly associated in mandated homeschoolers (both male and female parents; all *p*-values > 0.05).

## 4. Discussion

In the present study, we investigated whether equity and/or specialisation was associated with greater well-being among parents during the COVID-19 pandemic and tested whether these effects varied by gender and/or homeschooling status. It was predicted that parents with a highly specialised and/or equitable division of labour would report lower levels of depression, anxiety, stress, and/or relationship conflict, as well as higher levels of relationship satisfaction. We also predicted that these relations would be stronger in mothers vs. fathers and that mandated and voluntary homeschooling parents would benefit more from equity than specialisation, while parents of in-person learners would benefit from both equity and specialisation. Our predictions were partially supported by our data analyses.

In terms of investigating the impact of the division of labour on parent well-being overall, we found high levels of specialisation, but not equity, to have positive effects on parents’ emotional and relationship well-being. Consistent with our hypothesis **(H1)** and with economic theory [18], high levels of specialisation among couples with children living through the COVID-19 pandemic was associated with increased emotional and relationship well-being in parents (i.e., lower reported levels of depression, anxiety, perceived stress, and relationship conflict). Building on pre-pandemic research showing the benefits of specialisation [32], our results suggest that specialisation might be an effective division of labour strategy to promote parents’ well-being during the COVID-19 pandemic, overall. Interestingly, a more equitable division of labour was not associated with similar benefits on parental well-being. This is in contrast with our hypothesis **(H2)** and past research demonstrating the benefits of equity in a relationship [71,73]. One possibility is that couples with an inequitable division of labour may not perceive it as being unfair or inequitable. This is consistent with research that finds that majority of couples who report an inequitable division of household labour also perceive these household arrangements to be fair to themselves and their partners [130,131,132]. It is possible that couples who adopt more traditional gender role attitudes might be particularly more accustomed to expect and/or accept an inequitable division of labour [133] because performing gendered tasks (regardless of the time and effort it takes) provides individuals the satisfaction of presenting themselves as competent members of their gender category [134,135]. Overall, our results offer evidence In support of the economic theory of specialisation rather than the psychological theory of equity. However, these overall effects should be interpreted in conjunction with the moderating effects of gender and homeschooling status discussed next.

Inconsistent with our hypothesis **(H3)**, the specialised division of labour was associated with more positive impacts on emotional and relationship well-being in men relative to women. Specifically, men in relationships with more specialised strategies for dividing labour reported lower levels of depression, anxiety, stress, and relationship conflict. We did find some evidence to suggest that women might also experience decreased anxiety with more specialisation, although this effect was marginal and smaller in magnitude then in the men and there were no associations with reduced stress, depression, or relationship conflict (See Figure 1). The observation of greater beneficial effects for men is partially consistent with pre-pandemic findings that the division of labour has more negative mental health consequences for women relative to men [64,81]. Moreover, these gendered effects of specialisation favouring men are consistent with pre-pandemic findings that show that specialisation confers more financial and employment benefits for men relative to women [136]. Furthermore, research shows that compared to mothers, fathers attribute more social value to paid work [137,138] and report more guilt, failure, and depression when unemployed [138,139]. Accordingly, men may find more specialised strategies for dividing labour to be beneficial for their mental and relational health. In contrast, research suggests that mothers attribute more meaning and happiness with domestic chores, leisure activities, and childcare tasks [140] and experience more guilt, depression, and anxiety when they perceive their work to interfere with their ability to care for their children [141]. Accordingly, women may also find more specialised strategies for dividing labour to be beneficial for their mental and relational health; however, this was not evident in the current study. Specialisation was only marginally associated with decreased anxiety in women. A potential explanation for the gender difference that emerges in the effects of specialisation on the well-being of fathers vs. mothers might be the gender norms that permeate society. An individual’s perceptions of equity and/or fairness in the division of household labour is filtered through gender ideology, and in turn influence emotional and relationship well-being [74]. Although many stereotypes of the “typical woman” have evolved over the past decade, gender norms regarding men are more resistant to change [142]. Further, changes such as parenthood may cause couples to adopt and/or maintain traditional gender roles [143,144]. However, our measures did not ask about gender role attitudes, limiting our ability to firmly attribute these gender differences to gender ideology. Future studies should investigate mechanisms underlying *why* specialisation might confer mental health and relational benefits to men more so than to women during the pandemic. Further, studies should explore whether specialisation was enforced on women during the COVID-19 pandemic (i.e., due to job loss and workplace closures) and whether such circumstances might influence parental well-being.

Consistent with our hypothesis **(H4),** an equitable division of labour was associated with more positive impacts for women as compared to men’s well-being. Specifically, women in relationships with a more equitable division of labour reported marginally lower levels of depression and significantly higher levels of relationship satisfaction compared with women in less equitable arrangements. In contrast, an equitable division of labour was not significantly associated in either a beneficial or detrimental direction with men’s emotional and relationship well-being (see Figure 2). This is consistent with pre-pandemic research, which suggests that an inequitable division of labour has greater psychological and interpersonal consequences on women relative to men [64,71,72,81]. Despite evolving societal ideas around men’s and women’s work and family roles, mothers are still expected to take on more physically and emotionally taxing responsibilities (e.g., managing and overseeing the family and household schedule) compared to their partners [85]. Pre-pandemic evidence suggests that these additional responsibilities take a toll on the mental health of mothers [31,97,145,146], a finding that has been replicated in COVID-19 pandemic research [14,92]. Our study builds on current research to show that women’s greater emotional responsiveness to the inequitable division of labour persists during the COVID-19 pandemic. This is worrisome in the context of COVID-19 containment measures (e.g., social distancing, school closures), when many household and childcare responsibilities have increased (e.g., Carlson et al., 2020b; Craig & Churchill, 2021; Shafer et al., 2020). These increased demands and their consequent effects on mental health and relationship well-being are also likely to interfere with a mother’s ability to parent during this challenging time and place them at increased risk for escalating conflict and domestic violence [147].

Our findings also highlighted the role of homeschooling status as a potential moderator of relationships between the division of labour strategies and parental well-being. In contrast with our predictions **(H5)**, specialisation more so than equity emerged as a useful strategy, with beneficial effects on the emotional and relationship well-being of parents who are voluntarily homeschooling their children. Specifically, while higher levels of specialisation were associated with significantly less depression, anxiety, and relationship conflict and higher relationship satisfaction in these parents (See Figure 3), more equity was associated with marginally lower relationship satisfaction (see Figure 4). This is consistent with the voluntary homeschooling literature, which suggests that mothers are often largely responsible for homeschooling children alongside other domestic responsibilities, although fathers may less frequently take on the primary homeschooler role [93]. Specialisation likely promotes more positive interpersonal relationships (as compared to equity) because it is consistent with a lot of the characteristics of homeschooling families. For instance, homeschooling families often decide to live on one income [93] and thus might find it more efficient if only one member of the couple must balance work and family commitments.

In contrast, none of the aforementioned relations between equity/specialisation and parental well-being were significant for parents who were mandated to homeschool their children due to the COVID-19 pandemic (see Figure 3 and Figure 4). This is inconsistent with our predictions **(H5)** and recent findings highlighting the protective effect of an egalitarian division of homeschooling labour on drinking frequency in couples during COVID-19 [92]. A potential explanation for our findings might be that couples who have experienced more waves of the pandemic and subsequently more instances of school closures have adapted in other ways to the increased demands of homeschooling opportunities. The supports for mandated homeschoolers have also likely evolved relative to the first wave of the pandemic (e.g., increased hours of online learning offered by schools, more educational supports). Furthermore, our mandatory homeschooling sample was composed of families with relatively high SES (see Table 1). Accordingly, we are not able to capture impacts on families who might be most susceptible to the negative impacts of the pandemic [148,149] and are consequently more likely to benefit from a more equal division of labour during mandated homeschooling. Interestingly, however, both specialisation and equity appeared to promote parental well-being for parents whose children attend school in-person, though the former had a larger effect. Consistent with **H5**, parents of in-person learners had a marginally negative association between specialisation and parental depression and relationship conflict, and a marginally positive association between equity and relationship satisfaction (see Figure 4). These findings suggest that parents who do not homeschool are susceptible to the benefits of either approach to the division of labour. However, it is also important to be aware that all these relationships might be bi-directional. Accordingly, factors such as better emotional adjustment and more relationship satisfaction are likely to promote positive and more frequent discussions regarding the division of labour in the home. More longitudinal analyses in the future might unpack the direction of these relations.

However, it is also important to recognise and interpret results considering the 3-way interactions between gender, homeschooling status, and specialisation/equity that were revealed in our supplementary analyses (See Figure 5, Figure 6 and Figure 7). Specifically, our results indicate that among couples who were voluntarily homeschooling, fathers reported more benefits associated with greater role specialisation (i.e., significantly lower relationship conflict). In contrast, in couples whose children were attending school in-person, both mothers and fathers appeared to benefit from equity (i.e., marginally lower depression or relationship conflict) but not specialisation. Furthermore, there were no significant relations between either equity or specialisation and parental well-being among mothers or fathers of mandated homeschoolers or in mothers who were voluntarily homeschooling. To reconcile these opposing reports on the effects of equity and specialisation, parents might be advised to adopt combinations of both specialised and equitable divisions of labour. Research that views these concepts as mutually exclusive or that examines equity alone might fail to capture the distinct impacts of these two approaches. It is also important to note that although some of the simple slopes noted above (see Figure 5, Figure 6 and Figure 7) were not significant (and/or were approaching significance), these are likely due to the reduced power of the current sample to detect effects for these higher order interactions; future studies should attempt to see if these preliminary findings are replicated using a larger sample. Overall, this work extends pre-pandemic research on division of labour and well-being [27] to the context of the COVID-19 pandemic and clarifies the role that homeschooling status and gender may play in moderating these relations.

### Limitations and Future Directions

Our study was limited in several ways. First, given the difficulty of obtaining objective measures of the division of labour, we relied on more subjective reports. Accordingly, our reports of equity and specialisation may not be reflective of the actual objective division of labour in a relationship. Nonetheless, subjective perceptions of the distribution of work in a relationship have been found to have greater impacts on relationship well-being than the actual, objective division of labour [60,61]. Future work should attempt to look at whether these effects persist with more objective measures of division of labour. Second, we relied on retrospective reports of individuals. Accordingly, time delays, changes in COVID-19 containment measures, and changes in the division of labour might have influenced participants’ responses. Nonetheless, research shows that division of labour is resistant to short-term changes in relationships [150]. Additionally, COVID-19 containment measures were relatively stable over the course of January–February 2021, and parents’ experiences of these conditions (during the second wave of the pandemic) are crucial to document, despite the time delay between the experience and the report. Future work should attempt to examine whether effects persist over time and across future waves of the pandemic. Third, most of our sample was university educated, Caucasian, and in a mixed-gender relationship; we also only recruited couples. Thus, we are limited in our ability to generalise findings. Future work should attempt to recruit more socially diverse samples (i.e., single-parent families and individuals of different race, ethnicity, socioeconomic status, gender, and/or sexual orientation). This is vital given predictions that COVID-19 is exacerbating pre-existing inequalities, e.g., across income levels and ethnic backgrounds [148,149]. It is important to note though that our sample’s socioeconomic, ethnic, and familial compositions (e.g., families with same-gender parents) are generally representative of the North American population [151,152,153,154,155]. Furthermore, although we attempted to be more inclusive in our initial analyses (**H1**–**H2**; **H5**) by using indistinguishable dyads, our small sample of same-gender couples prevented more in-depth analyses of the potential role of sexual orientation. Future work should attempt to see if our findings can be replicated in same-gender couples, particularly in light of research suggesting that equity might be particularly beneficial in these couples [32,156].

Fourth, it is likely that children affect their parents’ mental health, particularly if they require more support during mandated homeschooling. We did not ask about children’s ability to cope with homeschooling; thus, we are limited in our ability to attribute our results on parental well-being to division of labour strategies. However, our analyses controlled for the number of children in the household, and we did restrict recruitment to school-aged children in grades 1–5, who are more likely to experience mental health problems and require support and direction during mandated homeschooling [157]. In addition, the average age of the children in our sample did not differ across the three groups (Table 1). Collectively, this increases the likelihood that the effects observed in our study are a consequence of the division of labour rather than due to group differences in a potentially confounding variable. Despite our attempts to control for confounding effects in several ways, future research should attempt to clarify these relationships by inquiring about children’s experiences during mandated homeschooling. Fifth, the current study conceptualised childcare, housework, and homeschooling all under the umbrella of unpaid labour, which prevented us from examining the effects of partners specialising in specific kinds of unpaid labour. The purpose of the current study was to explore the effects of equity and specialisation (as conceptualised in the literature) on parents’ well-being during the COVID-19 pandemic; thus, such types of more specific analyses are beyond the scope of the current study. Nonetheless, future studies should aim to develop techniques aimed at exploring the effects of specialising in specific kinds of unpaid labour.

Sixth, our use of the Qualtrics Survey Panels as a recruitment tool in addition to our supplementary efforts to recruit voluntary homeschoolers may limit our ability to generalise findings. However, a recent meta-analysis found that samples drawn from online panel data are comparable to conventionally sourced data with respect to both psychometric properties and criterion validity [158]. Nonetheless, future research should attempt to investigate these relations in other samples and be obtained using more conventional recruitment methods. Seventh, our study’s measure of relationship conflict was limited in its focus on perpetration of conflict rather than victimization by conflict perpetrated by the partner. Consequently, future studies should attempt to see if effects differ when examining both the perpetration of and victimization by conflict. Further, our two relationship outcomes showed differing effects. For instance, equity was best for enhancing relational satisfaction in women, whereas specialisation was best for minimizing conflict in men. This might reflect how different genders contextualise relationship well-being and the relative importance that they attribute to relationship satisfaction and/or conflict in a partnership; indeed, this presents a potential avenue for future research. Eighth, due to the small sample size of parents who reside in the US (*n* = 156) in this study, we were unable to examine differences in parents based on their country of residence, as such analyses were likely to be underpowered. Responses to the COVID-19 pandemic likely differed between Canada and the US (e.g., travel restrictions, government support, school closures, health system capacity), which may have had differing impacts on parents’ responsibilities and work–life arrangements [159]. Future work may wish to examine whether the COVID-19 pandemic has had different effects on the division of labour in parents living in Canada vs. the United States.

## 5. Conclusions

In a more general sense, our findings offer support for the effects of the economic theory of specialisation on couples’ wellbeing [18]. When contrasting the effects of equity and specialisation on parents’ emotional and relationship well-being, consistent with the economic theory, specialisation emerged as a more beneficial strategy for couples to adopt during the COVID-19 pandemic to promote their well-being. However, these effects should be interpreted in the context of the observed moderating effects of an individual’s gender and homeschooling status. Although more role specialisation was beneficial for both mothers and fathers’ well-being, particularly fathers, equity was only beneficial for mothers’ well-being. These relations were further distinguished by a parent’s homeschooling status such that the well-being of parents who were voluntarily homeschooling benefitted more from specialisation while the well-being of parents with children attending school in-person benefited from the use of either strategy; mandated homeschoolers’ well-being did not appear to strongly benefit from the use of either equity or specialisation strategies. These results suggest that mandated homeschoolers have likely resorted to other strategies and supports to be able to adapt in other ways to the increased demands of homeschooling opportunities during COVID-19; as such, these parents might require more external and educational supports to promote their well-being.

Overall, our findings advance the current literature on the division of labour and its impacts on emotional and relationship well-being by isolating the specific impacts of specialisation and equity and by examining the division of various forms of labour (paid and unpaid), including homeschooling labour. Furthermore, the large amount of homeschooling literature to date has studied almost exclusively voluntary contexts [160,161,162]. Moreover, the emerging COVID-19 pandemic-mandated homeschooling literature [14,92] has focused on comparing mandated homeschooling couples with couples without children. However, there might be important similarities and differences between mandated and voluntary homeschoolers. This study presents an incremental advance in the voluntary homeschooling and emerging COVID-19 mandatory homeschooling literature by being among the first to compare the outcomes of both mandated and voluntary homeschooling populations with those of in-person learners during the pandemic.

Practically speaking, our results inform our understanding of how different divisions of labour support (or interfere with) emotional well-being for parents impacted by COVID-19 pandemic-related disruptions. Indeed, approximately 4–8% of the variance in our well-being outcomes is accounted for by specialisation and/or equity, suggesting that they have meaningful impacts on well-being. This is vital information to have in the face of the current pandemic and the increased need for mental health services [163], especially as the division of labour might present itself as a potential therapeutic target for couples in therapy [164]. Division of labour is also uniquely suited as a target for interventions given its adaptability and malleable nature in comparison to other risk factors that might impact couples’ wellbeing, e.g., personality, genetics. For instance, clinicians might provide guidance on how to divide up work and family tasks in a relationship to avoid conflict, promote relationship satisfaction, and support the mental health of both members of a couple. Specifically, based on the results of this study, clinicians might advise couples who are voluntarily homeschooling to promote a more specialised division of labour strategies, and parents with children studying at in-person school might be best advised to adopt highly equitable and/or specialised division of labour arrangements (with potential additive effects of adopting both). However, when dealing with mixed-gender couples specifically, regardless of their homeschooling status, couples should be advised to attempt to adopt both equitable and specialised division of labour strategies to ensure positive impacts on both mothers’ and fathers’ emotional and relationship well-being.

## Figures and Tables

**Figure 1 ijerph-19-17021-f001:**
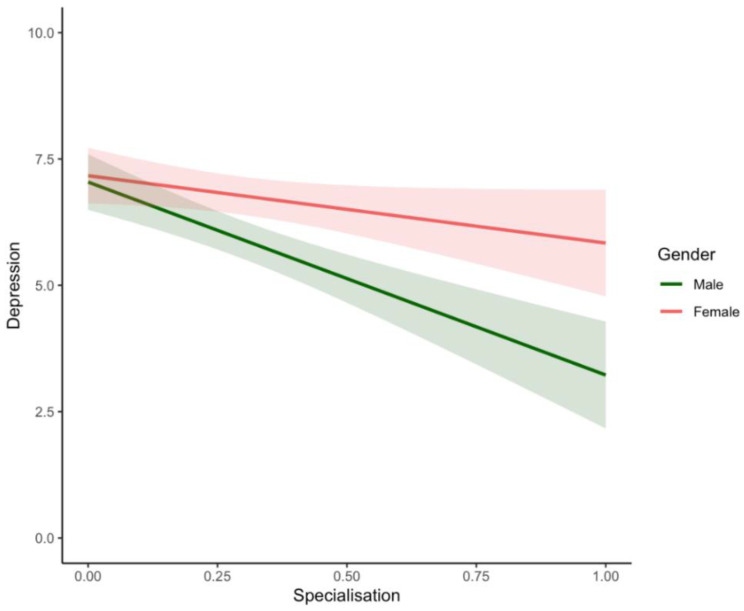
Simple slopes analyses demonstrating the interaction between specialisation and gender on parents’ depression in mixed-sex couples.

**Figure 2 ijerph-19-17021-f002:**
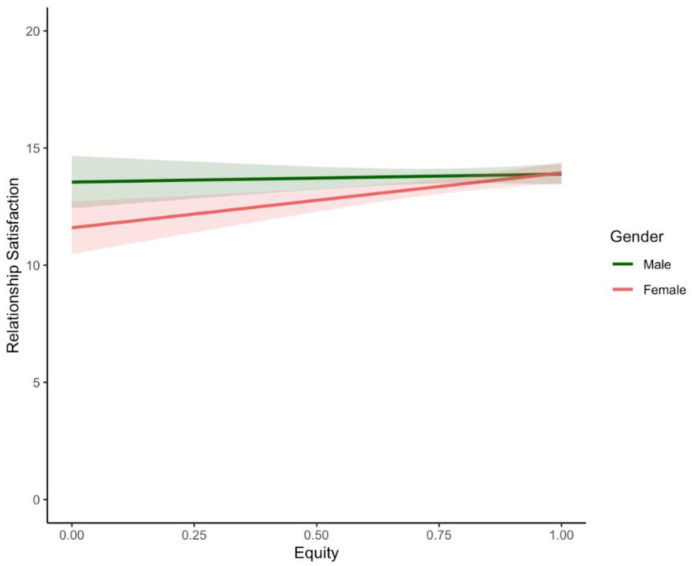
Simple slopes analyses demonstrating the interaction between equity and gender on parents’ relationship satisfaction in mixed-sex couples.

**Figure 3 ijerph-19-17021-f003:**
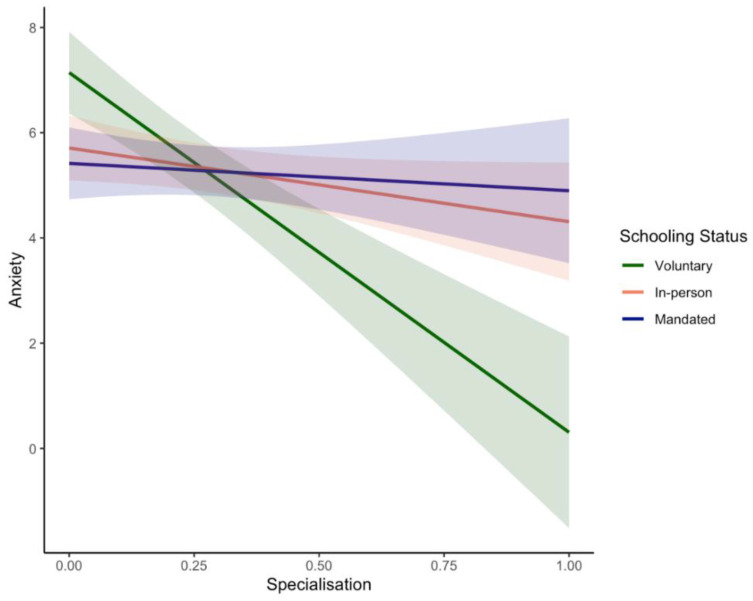
Simple slopes analyses demonstrating the interaction between specialisation and schooling status on parents’ anxiety.

**Figure 4 ijerph-19-17021-f004:**
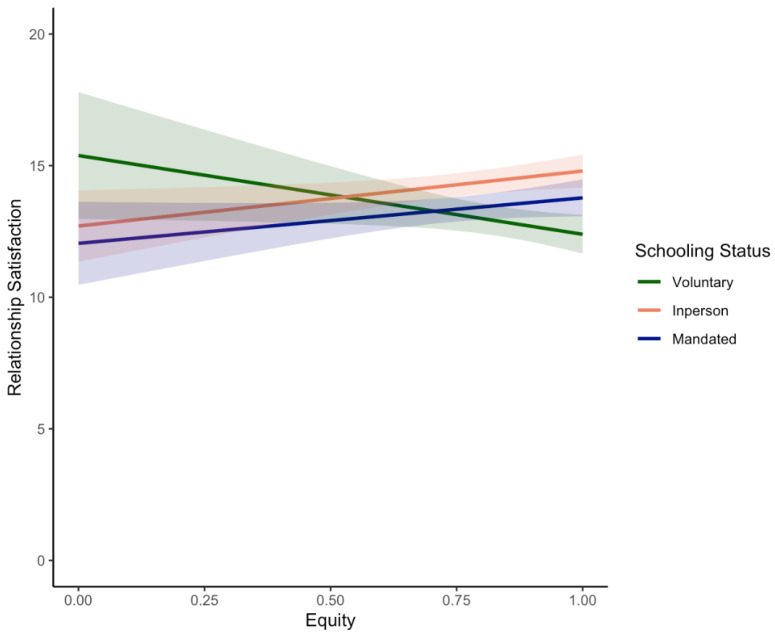
Simple slopes analyses demonstrating the interaction between equity and schooling status on parents’ relationship satisfaction.

**Figure 5 ijerph-19-17021-f005:**
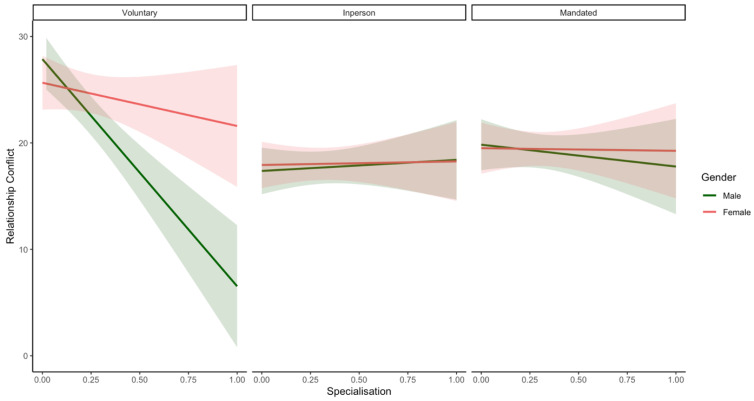
Simple slopes analyses demonstrating the three-way interaction between specialisation, gender, and schooling status on parents’ relationship conflict in mixed-gender couples.

**Figure 6 ijerph-19-17021-f006:**
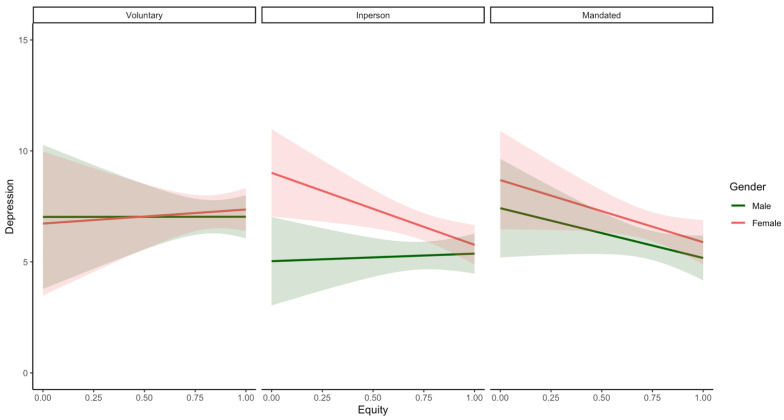
Simple slopes analyses demonstrating the three-way interaction between equity, gender, and schooling status on parents’ depression in mixed-gender couples.

**Figure 7 ijerph-19-17021-f007:**
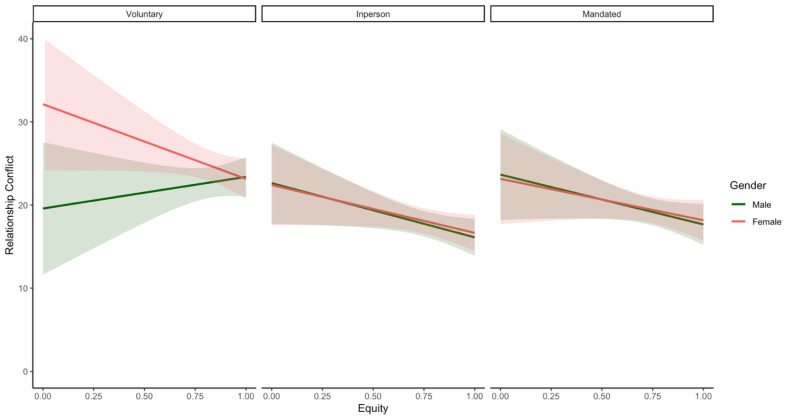
Simple slopes analyses demonstrating the three-way interaction between equity, gender, and schooling status on parents’ relationship conflict in mixed-gender couples.

**Table 1 ijerph-19-17021-t001:** Demographics information and relationship characteristics for the study sample.

Variable	In-Person Learners (*n* = 772)	Mandated Homeschoolers (*n =* 664)	Voluntary Homeschoolers (*n* = 488)
Age—M (SD) ***	39.13 (6.70)	38.70 (6.91)	36.17 (5.61)
Number of children—M (SD) ***	2.10 (0.85)	1.97 (0.85)	1.48 (0.88)
Average child age—M (SD) ^†^	8.64 (2.75)	8.80 (2.86)	8.44 (2.03)
Relationship length (yrs)—M (SD) ***	13.63 (6.24)	13.05 (6.19)	10.64 (5.05)
Gender			
Male	388	334	242
Female	383	328	244
Other/ Prefer not to answer	1	2	2
Relationship Status ***			
Mixed gender	642	562	458
Same gender	128	98	26
Other/Prefer not to answer	2	4	4
Family Income ***			
CAD 25,000 or less per year	36	33	8
Between CAD 26,000 and 50,000	78	76	83
Between CAD 51,000 and 75,000	98	110	160
Between CAD 76,000 and 100,000	172	106	122
Between CAD 101,000 and 125,000	110	84	60
Between CAD 126,000 and 150,000	117	94	38
CAD 151,000 or more per year	128	132	10
Prefer not to answer	33	29	7
Highest Level of Education ***			
Some high school	17	15	10
High school graduate	81	74	44
Some college/university	88	67	130
College/university graduate	414	301	214
Some post-graduate	41	34	57
Post-graduate degree	131	173	33
Employment ***			
Employed (Full/Part time)	634	529	389
Unemployed	116	129	89
Student (Full/Part time)	21	4	1
Prefer not to answer	1	2	9
Ethnicity ***			
White	552	438	416
Asian or Arab/West Asian ^1^	131	126	23
Latin American or Black or First Nations ^1^	52	60	37
Multiracial	25	28	6
Other/Unknown	12	12	6

*Notes*. ^1^ Combined to maintain confidentiality of respondents due to low numbers in one or more of these categories. ^†^ *p* < 0.10; *** *p* < 0.001.

**Table 2 ijerph-19-17021-t002:** Descriptive statistics and bivariate correlations among study variables.

	1	2	3	4	5	6	7	In-Person Learners—Mean (SD)	Mandated HS— Mean (SD)	Voluntary HS— Mean (SD)
Depression		0.83 **	0.56 **	−0.42 **	0.52 **	−0.08 ^†^	−0.06	6.19 (5.96)	6.29 (5.61)	7.41 (5.65)
2. Anxiety	0.86 **		0.58 **	−0.42 **	0.50 **	−0.07	−0.03	5.05 (5.11)	5.24 (4.96)	5.88 (4.78)
3. Perceived Stress	0.51 **	0.53 **		−0.46 **	0.39 **	−0.08 ^†^	−0.02	6.32 (2.87)	6.41 (3.10)	7.10 (2.46)
4. Relationship Satisfaction	−0.36 **	−0.35 **	−0.40 **		−0.55 **	0.07	−0.06	14.45 (4.48)	13.39 (4.79)	12.64 (4.27)
5. Relationship Conflict	0.55 **	0.50 **	0.38 **	−0.51 **		−0.07	−0.03	19.36 (14.07)	19.22 (13.46)	24.26 (15.38)
6. Equity	0.00	0.02	−0.05 ^†^	0.02	0.01		−0.18 **	0.74 (0.25)	0.75 (0.23)	0.84 (0.19)
7. Specialisation	−0.14 **	−0.13 **	−0.06 *	0.01	−0.12 **	−0.33 **		0.29 (0.30)	0.28 (0.26)	0.22 (0.24)
Mean (Total sample)	6.54	5.33	6.55	13.62	20.55	0.77	0.27			
SD (Total sample)	5.78	4.98	2.87	4.59	14.37	0.23	0.27			

*Notes.* Correlations for the whole sample (*n* = 1924 individuals; 962 couples) are below the diagonal; for the mandated homeschoolers (*n* = 664 individuals; 332 couples), the correlations are above the diagonal. HS stands for Homeschoolers. ^†^ *p* < 0.05; * *p* < 0.01; ** *p* < 0.001.

**Table 3 ijerph-19-17021-t003:** Effects of specialisation on parents’ well-being.

Outcome	Predictors	*B*	SE (*b*)	95% CI	*p*
**Depression** Marginal R^2^/Conditional R^2^ = 0.044/0.622 AIC/BIC = 11,303/11,353	Intercept	10.46 **	1.21	8.08–12.83	<0.001
	Parent age	−0.06	0.03	−0.12–0.00	0.068
	No. of children	−0.48 ^†^	0.2	−0.87–−0.09	0.015
	Relationship length	−0.04	0.04	−0.11–0.03	0.253
	Family Income	−0.07	0.1	−0.26–0.12	0.470
	Education	0.01	0.11	−0.20–0.22	0.927
	Specialisation	−2.91 **	0.62	−4.12–−1.69	<0.001
**Anxiety** Marginal R^2^/Conditional R^2^ = 0.036/0.544 AIC/BIC = 10,882/10,932	Intercept	8.60 **	1.03	6.57–10.62	<0.001
	Parent age	−0.04	0.03	−0.10–0.01	0.106
	No. of children	−0.34†	0.17	−0.66–−0.01	0.043
	Relationship length	−0.04	0.03	−0.10–0.01	0.135
	Family Income	−0.01	0.08	−0.17–0.15	0.878
	Education	−0.07	0.1	−0.26–0.12	0.497
	Specialisation	−2.20 **	0.52	−3.22–−1.17	<0.001
**Perceived Stress** Marginal R^2^/Conditional R^2^ = 0.030/0.495 AIC/BIC = 8878.0/8927.8	Intercept	9.22 **	0.59	8.07–10.36	<0.001
	Parent age	−0.04 *	0.02	−0.07–−0.01	0.007
	No. of children	−0.17	0.09	−0.35–0.02	0.076
	Relationship length	−0.001	0.02	−0.03–0.03	0.935
	Family Income	−0.14 *	0.05	−0.23–−0.05	0.003
	Education	−0.03	0.06	−0.14–0.08	0.559
	Specialisation	−0.68 ^†^	0.3	−1.26–−0.10	0.022
**Relationship Satisfaction** Marginal R^2^/Conditional R^2^ = 0.026/0.779 AIC/BIC = 10,054/10,103	Intercept	12.49 **	0.99	10.54–14.43	<0.001
	Parent age	−0.02	0.03	−0.08–0.03	0.362
	No. of children	0.76 **	0.17	0.44–1.08	<0.001
	Relationship length	0.04	0.03	−0.02–0.10	0.200
	Family Income	0.04	0.08	−0.11–0.19	0.625
	Education	−0.01	0.07	−0.16–0.14	0.901
	Specialisation	0.02	0.52	−1.01–1.04	0.972
**Relationship Conflict** Marginal R^2^/Conditional R^2^ = 0.043/0.703 AIC/BIC = 14,513/14,562	Intercept	30.33 **	3.06	24.34–36.33	<0.001
	Parent age	−0.15	0.08	−0.31–0.01	0.058
	No. of children	−1.58 *	0.5	−2.57–−0.59	0.002
	Relationship length	−0.15	0.09	−0.33–0.03	0.094
	Family Income	0.36	0.24	−0.11–0.83	0.138
	Education	−0.08	0.26	−0.58–0.42	0.749
	Specialisation	−5.07 *	1.59	−8.19–−1.96	0.001

*Note.* ^†^ *p* < 0.05; * *p* < 0.01; ** *p* < 0.001.

**Table 4 ijerph-19-17021-t004:** Effects of equity on parents’ well-being.

Outcome	Predictors	*B*	SE (*b*)	95% CI	*p*
**Depression** Marginal R^2^/Conditional R^2^ = 0.026/0.622 AIC/BIC =11,324/11,374.2	Intercept	10.98 **	1.22	8.59–13.37	<0.001
	Parent age	−0.06	0.03	−0.12–0.00	0.055
	No. of children	−0.65 *	0.2	−1.04–−0.26	0.001
	Relationship Length	−0.05	0.04	−0.12–0.02	0.153
	Family Income	−0.05	0.1	−0.24–0.14	0.602
	Education	0.01	0.11	−0.21–0.22	0.961
	Equity	−0.34	0.74	−1.79–1.10	0.642
**Anxiety** Marginal R^2^/Conditional R^2^ = 0.022/0.544 AIC/BIC =10,899/10,949.0	Intercept	8.99 **	1.04	6.96–11.02	<0.001
	Parent age	−0.05	0.03	−0.10–0.01	0.090
	No. of children	−0.45 *	0.17	−0.78–−0.12	0.007
	Relationship Length	−0.05	0.03	−0.11–0.01	0.082
	Family Income	−0.01	0.08	−0.17–0.16	0.941
	Education	−0.07	0.1	−0.26–0.12	0.469
	Equity	0.12	0.62	−1.10–1.34	0.848
**Perceived Stress** Marginal R^2^/Conditional R^2^ = 0.029/0.495 AIC/BIC = 8880/8929.7	Intercept	9.34 **	0.58	8.19–10.48	<0.001
	Parent age	−0.04 *	0.02	−0.07–−0.01	0.006
	No. of children	−0.23 ^†^	0.09	−0.41–−0.04	0.015
	Relationship Length	0.001	0.02	−0.04–0.03	0.789
	Family Income	−0.12 *	0.05	−0.22–−0.03	0.009
	Education	−0.03	0.06	−0.14–0.08	0.571
	Equity	−0.63	0.35	−1.31–0.05	0.070
**Relationship Satisfaction** Marginal R^2^/Conditional R^2^ = 0.027/0.779 AIC/BIC = 10,052/10,101.7	Intercept	12.47 **	0.98	10.54–14.40	<0.001
	Parent age	−0.02	0.03	−0.08–0.03	0.369
	No. of children	0.79 **	0.17	0.47–1.12	<0.001
	Relationship Length	0.04	0.03	−0.02–0.10	0.189
	Family Income	0.02	0.08	−0.13–0.17	0.760
	Education	−0.01	0.07	−0.16–0.13	0.879
	Equity	0.77	0.61	−0.43–1.97	0.208
**Relationship Conflict** Marginal R^2^/Conditional R^2^ = 0.034/0.703 AIC/BIC = 14,523/14,572.4	Intercept	31.25 **	3.06	25.25–37.24	<0.001
	Parent age	−0.16	0.08	−0.32–−0.00	0.050
	No. of children	−1.88 **	0.5	−2.87–−0.90	<0.001
	Relationship Length	−0.17	0.09	−0.35–0.01	0.061
	Family Income	0.39	0.24	−0.09–0.87	0.107
	Education	−0.09	0.26	−0.59–0.41	0.732
	Equity	−0.94	1.87	−4.61–2.73	0.616

*Note.* ^†^ *p* < 0.05; * *p* < 0.01; ** *p* < 0.001.

**Table 5 ijerph-19-17021-t005:** Effects of specialisation on Male vs. Female parents’ well-being in mixed-sex couples.

Outcome	Predictors	*B*	SE (*b*)	95% CI	*p*
**Depression** Marginal R^2^/Conditional R^2^ = 0.054/0.598 AIC/BIC = 9669.8/9729.0	Intercept	10.39 **	1.24	7.96–12.81	<0.001
	Parent age	−0.05	0.03	−0.11–0.01	0.127
	No. of children	−0.52 *	0.2	−0.91–−0.14	0.008
	Relationship length	−0.04	0.04	−0.11–0.03	0.283
	Family Income	−0.12	0.1	−0.31–0.07	0.229
	Education	−0.02	0.11	−0.24–0.20	0.873
	Specialisation	−2.58 **	0.63	−3.81–−1.34	<0.001
	Gender	0.40 **	0.09	0.22–0.57	<0.001
	Specialisation × Gender	1.24 **	0.32	0.62–1.87	<0.001
**Anxiety** Marginal R^2^/Conditional R^2^ = 0.050/0.532 AIC/BIC = 9345.8/9405.0	Intercept	8.62 **	1.07	6.51–10.72	<0.001
	Parent age	−0.04	0.03	−0.09–0.02	0.166
	No. of children	−0.4 ^†^	0.17	−0.74–−0.07	0.018
	Relationship length	−0.04	0.03	−0.10–0.02	0.199
	Family Income	−0.04	0.09	−0.21–0.13	0.614
	Education	−0.07	0.1	−0.27–0.13	0.468
	Specialisation	−2.12 **	0.54	−3.19–−1.06	<0.001
	Gender	0.45 **	0.08	0.28–0.61	<0.001
	Specialisation × Gender	0.87 *	0.3	0.27–1.46	0.004
**Perceived Stress** Marginal R^2^/Conditional R^2^ = 0.047/0.512 AIC/BIC = 7708.7/7767.8	Intercept	9.52 **	0.64	8.26–10.77	<0.001
	Parent age	−0.04 *	0.02	−0.08–−0.01	0.008
	No. of children	−0.19	0.1	−0.39–0.01	0.057
	Relationship length	0.001	0.02	−0.04–0.04	0.982
	Family Income	−0.16 *	0.05	−0.26–−0.06	0.001
	Education	−0.05	0.06	−0.17–0.07	0.430
	Specialisation	−0.55	0.32	−1.18–0.09	0.090
	Gender	0.27 **	0.05	0.17–0.37	<0.001
	Specialisation × Gender	0.56 *	0.18	0.20–0.92	0.002
**Relationship Satisfaction** Marginal R^2^/Conditional R^2^ = 0.033/0.788 AIC/BIC = 8646.9/8706.1	Intercept	12.57 **	1.05	10.50–14.64	<0.001
	Parent age	−0.02	0.03	−0.08–0.03	0.426
	No. of children	0.82 **	0.17	0.48–1.16	<0.001
	Relationship length	0.04	0.03	−0.03–0.10	0.236
	Family Income	0.02	0.08	−0.14–0.18	0.811
	Education	−0.04	0.08	−0.19–0.11	0.599
	Specialisation	−0.26	0.56	−1.35–0.83	0.641
	Gender	−0.19 **	0.05	−0.29–−0.09	<0.001
	Specialisation × Gender	−0.64 **	0.19	−1.02–−0.27	0.001
**Relationship Conflict** Marginal R^2^/Conditional R^2^ = 0.043/0.667 AIC/BIC = 12,483.8/12,543.0	Intercept	30.08 **	3.12	23.96–36.19	<0.001
	Parent age	−0.12	0.08	−0.28–0.04	0.156
	No. of children	−1.76 **	0.5	−2.74–−0.77	<0.001
	Relationship length	−0.18	0.09	−0.36–0.01	0.059
	Family Income	0.19	0.25	−0.30–0.67	0.445
	Education	−0.15	0.27	−0.67–0.38	0.584
	Specialisation	−3.26 ^†^	1.61	−6.41–−0.11	0.042
	Gender	0.3	0.2	−0.09–0.70	0.129
	Specialisation × Gender	1.91 *	0.72	0.49–3.32	0.008

*Note.* ^†^ *p* < 0.05; * *p* < 0.01; ** *p* < 0.001. Gender-coded −1 (Male), +1 (Female).

**Table 6 ijerph-19-17021-t006:** Effects of equity on Male vs. Female parents’ well-being in mixed-sex couples.

Outcome	Predictors	*B*	SE (*b*)	95% CI	*p*
**Depression** Marginal R^2^/Conditional R^2^ = 0.039/0.593 AIC/BIC = 9693.5/9752.7	Intercept	10.97 **	1.24	8.54–13.40	<0.001
	Parent age	−0.05	0.03	−0.12–0.01	0.101
	No. of children	−0.73 **	0.2	−1.12–−0.34	<0.001
	Relationship length	−0.05	0.04	−0.12–0.03	0.210
	Family Income	−0.08	0.1	−0.28–0.11	0.411
	Education	−0.05	0.11	−0.27–0.17	0.675
	Equity	−1.04	0.77	−2.55–0.47	0.177
	Gender	0.42 **	0.09	0.25–0.60	<0.001
	Equity × Gender	−0.96 ^†^	0.39	−1.72–−0.19	0.014
**Anxiety** Marginal R^2^/Conditional R^2^ = 0.036/0.531 AIC/BIC = 9363.3/9422.5	Intercept	9.07 **	1.08	6.96–11.19	<0.001
	Parent age	−0.04	0.03	−0.10–0.01	0.140
	No. of children	−0.55 *	0.17	−0.88–−0.21	0.001
	Relationship length	−0.05	0.03	−0.11–0.02	0.144
	Family Income	−0.03	0.09	−0.20–0.15	0.760
	Education	−0.1	0.1	−0.30–0.10	0.346
	Equity	−0.28	0.67	−1.59–1.02	0.670
	Gender	0.46 **	0.08	0.30–0.63	<0.001
	Equity × Gender	−0.88 ^†^	0.37	−1.60–−0.16	0.017
**Perceived Stress** Marginal R^2^/Conditional R^2^ = 0.045/0.508 AIC/BIC = 7714.6/7773.8	Intercept	9.66 **	0.64	8.42–10.91	<0.001
	Parent age	−0.05 *	0.02	−0.08–−0.01	0.006
	No. of children	−0.26 ^†^	0.1	−0.45–−0.06	0.010
	Relationship length	0.001	0.02	−0.04–0.03	0.925
	Family Income	−0.14 *	0.05	−0.25–−0.04	0.005
	Education	−0.06	0.06	−0.18–0.06	0.355
	Equity	−0.71	0.39	−1.48–0.06	0.069
	Gender	0.28 **	0.05	0.18–0.38	<0.001
	Equity × Gender	−0.37	0.22	−0.81–0.07	0.099
**Relationship Satisfaction** Marginal R^2^/Conditional R^2^ = 0.038/0.789 AIC/BIC =8635.5/ 8694.7	Intercept	12.45 **	1.05	10.40–14.50	<0.001
	Parent age	−0.02	0.03	−0.08–0.03	0.459
	No. of children	0.87 **	0.17	0.53–1.21	<0.001
	Relationship length	0.04	0.03	−0.03–0.10	0.253
	Family Income	−0.01	0.08	−0.17–0.15	0.859
	Education	−0.02	0.08	−0.17–0.13	0.830
	Equity	1.34 ^†^	0.67	0.03–2.66	0.046
	Gender	−0.20 **	0.05	−0.31–−0.10	<0.001
	Equity × Gender	1.00 **	0.23	0.55–1.46	<0.001
**Relationship Conflict** Marginal R^2^/Conditional R^2^ = 0.041/0.664 AIC/BIC = 12,491.6/ 12,550.8	Intercept	30.86 **	3.11	24.77–36.96	<0.001
	Parent age	−0.12	0.08	−0.29–0.04	0.133
	No. of children	−2.10 **	0.5	−3.09–−1.12	<0.001
	Relationship length	−0.18 ^†^	0.09	−0.37–−0.00	0.048
	Family Income	0.27	0.25	−0.22–0.76	0.278
	Education	−0.18	0.27	−0.70–0.34	0.501
	Equity	−3.21	1.95	−7.04–0.62	0.100
	Gender	0.34	0.2	−0.05–0.73	0.089
	Equity × Gender	−0.65	0.88	−2.38–1.07	0.458

*Note.* ^†^ *p* < 0.05; * *p* < 0.01; ** *p* < 0.001. Gender-coded −1 (Male), +1 (Female).

**Table 7 ijerph-19-17021-t007:** Effects of specialisation and homeschooling status on parents’ well-being.

Outcome	Predictors	*B*	SE (*b*)	95% CI	*p*
**Depression** Marginal R^2^/Conditional R^2^ = 0.054/0.623 AIC/BIC = 11,291.4/11,363.2	Intercept	10.20 **	1.38	7.49–12.91	<0.001
	Parent age	−0.05	0.03	−0.11–0.01	0.116
	No. of children	−0.37	0.2	−0.77–0.03	0.069
	Relationship Length	−0.04	0.04	−0.11–0.03	0.240
	Family Income	−0.08	0.1	−0.27–0.11	0.433
	Education	0.03	0.11	−0.18–0.25	0.760
	Specialisation	−12.33 **	3.53	−19.26–−5.41	<0.001
	Homeschooling 1	−0.01	0.24	−0.47–0.46	0.979
	Homeschooling 2	−0.18	0.27	−0.72–0.36	0.511
	Specialisation × Homeschooling 1	1.11	0.83	−0.52–2.73	0.182
	Specialisation × Homeschooling 2	3.32 *	1.02	1.33–5.32	0.001
**Anxiety** Marginal R^2^/Conditional R^2^ = 0.053/0.546 AIC/BIC = 10,866.6/10,938.4	Intercept	7.99 **	1.17	5.70–10.28	<0.001
	Parent age	−0.04	0.03	−0.09–0.02	0.183
	No. of children	−0.24	0.17	−0.57–0.10	0.169
	Relationship Length	−0.05	0.03	−0.11–0.01	0.112
	Family Income	−0.03	0.08	−0.20–0.13	0.686
	Education	−0.04	0.1	−0.23–0.15	0.648
	Specialisation	−13.79 **	2.97	−19.61–−7.97	<0.001
	Homeschooling 1	0.03	0.2	−0.36–0.42	0.885
	Homeschooling 2	0.003	0.23	−0.45–0.45	0.989
	Specialisation × Homeschooling 1	1.52 ^†^	0.7	0.15–2.88	0.029
	Specialisation × Homeschooling 2	3.92 **	0.85	2.24–5.59	<0.001
**Perceived Stress** Marginal R^2^/Conditional R^2^ = 0.034/0.496 AIC/BIC = 8879.1/8951.0	Intercept	9.61 **	0.67	8.30–10.92	<0.001
	Parent age	−0.04 ^†^	0.02	−0.07–−0.01	0.012
	No. of children	−0.12	0.1	−0.31–0.07	0.210
	Relationship Length	−0.000	0.02	−0.03–0.03	10.000
	Family Income	−0.13 *	0.05	−0.22–−0.03	0.008
	Education	−0.02	0.06	−0.13–0.09	0.683
	Specialisation	−0.88	1.69	−4.19–2.43	0.603
	Homeschooling 1	−0.09	0.11	−0.31–0.13	0.436
	Homeschooling 2	−0.25	0.13	−0.51–0.00	0.052
	Specialisation × Homeschooling 1	−0.11	0.4	−0.89–0.66	0.776
	Specialisation × Homeschooling 2	0.22	0.49	−0.73–1.17	0.651
**Relationship** **Satisfaction** Marginal R^2^/Conditional R^2^ = 0.049/0.780 AIC/BIC = 10,036.6/10,108.4	Intercept	10.93 **	1.13	8.71–13.15	<0.001
	Parent age	−0.04	0.03	−0.09–0.02	0.173
	No. of children	0.57 **	0.17	0.24–0.90	0.001
	Relationship Length	0.04	0.03	−0.02–0.09	0.226
	Family Income	0.02	0.08	−0.13–0.17	0.815
	Education	−0.02	0.07	−0.16–0.13	0.832
	Specialisation	8.42 *	2.95	2.64–14.20	0.004
	Homeschooling 1	0.73 **	0.2	0.34–1.11	<0.001
	Homeschooling 2	0.51 ^†^	0.23	0.07–0.96	0.024
	Specialisation × Homeschooling 1	−1.18	0.69	−2.53–0.18	0.088
	Specialisation × Homeschooling 2	−2.80 **	0.85	−4.46–−1.13	0.001
**Relationship Conflict** Marginal R^2^/Conditional R^2^ = 0.060/0.704 AIC/BIC = 14,498.5/14,570.4	Intercept	33.55 **	3.49	26.71–40.39	<0.001
	Parent age	−0.12	0.08	−0.28–0.04	0.139
	No. of children	−1.05†	0.52	−2.07–−0.04	0.042
	Relationship Length	−0.14	0.09	−0.32–0.03	0.114
	Family Income	0.46	0.24	−0.02–0.93	0.060
	Education	−0.01	0.26	−0.50–0.50	0.995
	Specialisation	−21.41 ^†^	8.99	−39.04–−3.78	0.017
	Homeschooling 1	−0.86	0.6	−2.04–0.32	0.153
	Homeschooling 2	−2.29 **	0.7	−3.66–−0.93	0.001
	Specialisation × Homeschooling 1	1.27	2.11	−2.87–5.40	0.548
	Specialisation × Homeschooling 2	6.52†	2.59	1.44–11.59	0.012

*Notes.*^†^ *p* < 0.05; * *p* < 0.01; ** *p* < 0.001. Homeschooling status 1 coded as 0 (voluntary homeschoolers), 1 (not homeschooling, i.e., child attending school in-person), and −1 (mandated homeschooling due to COVID-19). Homeschooling status 2 coded as −1 (voluntary homeschoolers), 0 (not homeschooling, i.e., child attending school in-person), and 1 (mandated homeschooling due to COVID-19).

**Table 8 ijerph-19-17021-t008:** Effects of equity and homeschooling status on parents’ well-being.

Outcome	Predictors	*B*	SE (*b*)	95% CI	*p*
**Depression** Marginal R^2^/Conditional R^2^ = 0.029/0.623 AIC/BIC = 11,328.5/11,400.3	Intercept	11.23 **	1.42	8.44–14.02	<0.001
	Parent age	−0.06	0.03	−0.12–0.01	0.078
	No. of children	−0.59 *	0.2	−0.99–−0.19	0.004
	Relationship Length	−0.05	0.04	−0.12–0.02	0.184
	Family Income	−0.02	0.1	−0.22–0.17	0.815
	Education	0.03	0.11	−0.19–0.24	0.815
	Equity	0.33	4.45	−8.39–9.06	0.941
	Homeschooling 1	−0.06	0.24	−0.54–0.41	0.792
	Homeschooling 2	−0.33	0.29	−0.89–0.23	0.248
	Equity × Homeschooling 1	0.58	1.01	−1.39–2.56	0.561
	Equity × Homeschooling 2	−0.97	1.28	−3.48–1.54	0.449
**Anxiety** Marginal R^2^/Conditional R^2^ = 0.025/0.546 AIC/BIC = 10,903.5/10,975.3	Intercept	8.78 **	1.21	6.41–11.14	<0.001
	Parent age	−0.04	0.03	−0.10–0.01	0.116
	No. of children	−0.42 ^†^	0.17	−0.75–−0.08	0.016
	Relationship Length	−0.05	0.03	−0.11–0.01	0.102
	Family Income	−0.01	0.09	−0.16–0.17	0.955
	Education	−0.06	0.1	−0.25–0.14	0.574
	Equity	3.01	3.75	−4.34–10.35	0.422
	Homeschooling 1	0	0.2	−0.40–0.40	0.988
	Homeschooling 2	−0.09	0.24	−0.56–0.39	0.723
	Equity × Homeschooling 1	0.14	0.85	−1.52–1.80	0.871
	Equity × Homeschooling 2	−1.5	1.08	−3.61–0.61	0.164
**Perceived Stress** Marginal R^2^/Conditional R^2^ = 0.035/0.496 AIC/BIC = 8879.9/8951.7	Intercept	9.71 **	0.68	8.38–11.04	<0.001
	Parent age	−0.04 ^†^	0.02	−0.07–−0.01	0.013
	No. of children	−0.17	0.1	−0.36–0.02	0.083
	Relationship Length	0.001	0.02	−0.04–0.03	0.896
	Family Income	−0.1 ^†^	0.05	−0.20–−0.01	0.033
	Education	−0.02	0.06	−0.13–0.09	0.735
	Equity	1.96	2.09	−2.14–6.06	0.350
	Homeschooling 1	−0.12	0.11	−0.34–0.11	0.310
	Homeschooling 2	−0.29 ^†^	0.13	−0.56–−0.03	0.030
	Equity × Homeschooling 1	−0.5	0.47	−1.43–0.43	0.290
	Equity × Homeschooling 2	−0.77	0.6	−1.95–0.41	0.199
**Relationship** **Satisfaction** Marginal R^2^/Conditional R^2^ = 0.051/0.780 AIC/BIC = 10,033.7/10,105.5	Intercept	11.06 **	1.15	8.81–13.31	<0.001
	Parent age	−0.03	0.03	−0.09–0.02	0.185
	No. of children	0.61 **	0.17	0.28–0.94	<0.001
	Relationship Length	0.03	0.03	−0.03–0.09	0.301
	Family Income	−0.02	0.08	−0.17–0.14	0.841
	Education	−0.02	0.07	−0.16–0.13	0.818
	Equity	−9.89 *	3.65	−17.05–−2.73	0.007
	Homeschooling 1	0.72 **	0.2	0.33–1.11	<0.001
	Homeschooling 2	0.51 ^†^	0.23	0.05–0.97	0.029
	Equity × Homeschooling 1	1.81 ^†^	0.83	0.20–3.43	0.028
	Equity × Homeschooling 2	3.27 *	1.05	1.21–5.33	0.002
**Relationship Conflict** Marginal R^2^/Conditional R^2^ = 0.048/0.704 AIC/BIC = 14,514.7/14,586.5	Intercept	35.69 **	3.56	28.71–42.68	<0.001
	Parent age	−0.13	0.08	−0.29–0.03	0.106
	No. of children	−1.47 *	0.52	−2.48–−0.46	0.004
	Relationship Length	−0.15	0.09	−0.33–0.02	0.089
	Family Income	0.58 ^†^	0.25	0.10–1.07	0.019
	Education	−0.01	0.26	−0.51–0.49	0.967
	Equity	−2.45	11.24	−24.47–19.57	0.827
	Homeschooling 1	−1.03	0.61	−2.23–0.17	0.091
	Homeschooling 2	−2.69 **	0.72	−4.10–−1.27	<0.001
	Equity × Homeschooling 1	1.33	2.54	−3.64–6.31	0.599
	Equity × Homeschooling 2	−1.21	3.23	−7.55–5.12	0.707

*Notes.*^†^ *p* < 0.05; * *p* < 0.01; ** *p* < 0.001. Homeschooling status 1 coded as 0 (voluntary homeschoolers), 1 (not homeschooling, i.e., child attending school in-person), and −1 (mandated homeschooling due to COVID-19). Homeschooling status 2 coded as −1 (voluntary homeschoolers), 0 (not homeschooling, i.e., child attending school in-person), and 1 (mandated homeschooling due to COVID-19).

**Table 9 ijerph-19-17021-t009:** Effects of specialisation and homeschooling status on male vs. female parents’ well-being in mixed-sex couples.

Outcome	Predictors	*B*	SE (*b*)	95% CI	*p*
**Depression** Marginal R^2^/Conditional R^2^ = 0.078/0.601 AIC/BIC = 9655.9/9758.1	Intercept	11.55 ***	1.32	8.97–14.13	<0.001
	Parent age	−0.05	0.03	−0.10–−0.00	0.050
	No. of children	−0.31	0.2	−0.70–0.09	0.125
	Relationship length	0.001	0.002	−0.00–0.00	0.433
	Family Income	−0.11	0.1	−0.31–0.08	0.247
	Education	−0.001	0.11	−0.22–0.22	0.969
	Specialisation	−16.43 ***	3.44	−23.18–−9.69	<0.001
	Gender	−0.55	0.46	−1.46–0.35	0.232
	Homeschooling 1	−0.46	0.24	−0.93–0.02	0.060
	Homeschooling 2	−0.61 ^†^	0.27	−1.15–−0.08	0.024
	Gender × Specialisation	0.28	1.77	−3.19–3.75	0.876
	Gender × Homeschooling 1	0.23	0.12	−0.01–0.47	0.060
	Gender × Homeschooling 2	0.23	0.13	−0.03–0.49	0.081
	Specialisation × Homeschooling 1	2.34 **	0.83	0.71–3.97	0.005
	Specialisation × Homeschooling 2	4.21 ***	0.99	2.27–6.15	<0.001
	Specialisation × Gender × Homeschooling 1	−0.04	0.43	−0.89–0.81	0.935
	Specialisation × Gender × Homeschooling 2	0.46	0.51	−0.54–1.46	0.369
**Anxiety** Marginal R^2^/Conditional R^2^ = 0.077/0.538 AIC/BIC = 9324.5/9426.7	Intercept	9.01 ***	1.14	6.77–11.25	<0.001
	Parent age	−0.05 ^†^	0.02	−0.09–−0.00	0.036
	No. of children	−0.28	0.17	−0.61–0.06	0.109
	Relationship length	0.001	0.001	−0.00–0.00	0.569
	Family Income	−0.07	0.09	−0.24–0.10	0.418
	Education	−0.07	0.1	−0.27–0.13	0.482
	Specialisation	−16.18 ***	2.96	−21.98–−10.37	<0.001
	Gender	−0.87 ^†^	0.44	−1.73–−0.01	0.047
	Homeschooling 1	−0.22	0.21	−0.63–0.18	0.282
	Homeschooling 2	−0.18	0.23	−0.63–0.28	0.454
	Gender × Specialisation	−0.48	1.67	−3.76–2.80	0.773
	Gender × Homeschooling 1	0.31 **	0.12	0.08–0.53	0.008
	Gender × Homeschooling 2	0.33 **	0.13	0.09–0.58	0.008
	Specialisation × Homeschooling 1	2.34 **	0.72	0.93–3.74	0.001
	Specialisation × Homeschooling 2	4.29 ***	0.85	2.62–5.95	<0.001
	Specialisation × Gender × Homeschooling 1	0.11	0.41	−0.69–0.92	0.782
	Specialisation × Gender × Homeschooling 2	0.48	0.48	−0.47–1.42	0.327
**Perceived Stress** Marginal R^2^/Conditional R^2^ = 0.052/0.515 AIC/BIC = 7715.8/7818.0	Intercept	9.93 ***	0.69	8.58–11.27	<0.001
	Parent age	−0.04 **	0.01	−0.07–−0.01	0.003
	No. of children	−0.13	0.1	−0.33–0.08	0.219
	Relationship length	0.001	0.001	−0.00–0.00	0.390
	Family Income	−0.15 **	0.05	−0.25–−0.04	0.005
	Education	−0.04	0.06	−0.16–0.08	0.520
	Specialisation	−1.26	1.78	−4.74–2.23	0.480
	Gender	0.10	0.27	−0.42–0.62	0.708
	Homeschooling 1	−0.11	0.13	−0.35–0.14	0.382
	Homeschooling 2	−0.29 ^†^	0.14	−0.57–−0.01	0.039
	Gender × Specialisation	1.63	1.02	−0.38–3.63	0.112
	Gender × Homeschooling 1	0.03	0.07	−0.11–0.17	0.676
	Gender × Homeschooling 2	0.05	0.08	−0.10–0.20	0.476
	Specialisation × Homeschooling 1	0.06	0.43	−0.79–0.90	0.897
	Specialisation × Homeschooling 2	0.29	0.51	−0.71–1.30	0.564
	Specialisation × Gender × Homeschooling 1	−0.43	0.25	−0.92–0.06	0.087
	Specialisation × Gender × Homeschooling 2	−0.08	0.3	−0.66–0.50	0.786
**Relationship Satisfaction** Marginal R^2^/Conditional R^2^ = 0.061/0.789 AIC/BIC = 8631.5/8733.7	Intercept	10.49 ***	1.13	8.28–12.71	<0.001
	Parent age	−0.02	0.02	−0.06–0.03	0.415
	No. of children	0.62 ***	0.18	0.27–0.97	<0.001
	Relationship length	−0.002	0.001	−0.00–0.00	0.192
	Family Income	0.00	0.08	−0.16–0.16	0.979
	Education	−0.04	0.08	−0.19–0.12	0.649
	Specialisation	8.69 **	3.04	2.74–14.64	0.004
	Gender	0.40	0.28	−0.14–0.95	0.144
	Homeschooling 1	0.77 ***	0.21	0.36–1.19	<0.001
	Homeschooling 2	0.66 **	0.24	0.19–1.13	0.006
	Gender × Specialisation	0.43	1.06	−1.64–2.50	0.683
	Gender × Homeschooling 1	−0.14	0.07	−0.29–0.00	0.051
	Gender × Homeschooling 2	−0.14	0.08	−0.30–0.01	0.068
	Specialisation × Homeschooling 1	−1.27	0.73	−2.71–0.17	0.084
	Specialisation × Homeschooling 2	−2.98 **	0.87	−4.69–−1.27	0.001
	Specialisation × Gender × Homeschooling 1	−0.15	0.26	−0.65–0.36	0.570
	Specialisation × Gender × Homeschooling 2	−0.34	0.31	−0.94–0.26	0.267
**Relationship Conflict** Marginal R^2^/Conditional R^2^ = 0.083/0.679 AIC/BIC = 12,435.3/12,537.5	Intercept	37.55 ***	3.31	31.05–44.04	<0.001
	Parent age	−0.14 ^†^	0.07	−0.27–−0.01	0.038
	No. of children	−1.01 ^†^	0.51	−2.01–−0.01	0.047
	Relationship length	0.002	0.004	−0.01–0.01	0.592
	Family Income	0.29	0.25	−0.20–0.77	0.243
	Education	−0.02	0.26	−0.54–0.50	0.945
	Specialisation	−30.63 ***	8.75	−47.77–−13.49	<0.001
	Gender	2.56 ^†^	1.03	0.54–4.58	0.013
	Homeschooling 1	−2.40 ***	0.61	−3.61–−1.20	<0.001
	Homeschooling 2	−3.24 ***	0.69	−4.59–−1.89	<0.001
	Gender × Specialisation	21.05 ***	3.94	13.33–28.77	<0.001
	Gender × Homeschooling 1	−0.31	0.27	−0.84–0.23	0.258
	Gender × Homeschooling 2	−0.72 ^†^	0.29	−1.30–−0.15	0.014
	Specialisation × Homeschooling 1	4.89 ^†^	2.11	0.75–9.04	0.021
	Specialisation × Homeschooling 2	8.16 **	2.51	3.24–13.09	0.001
	Specialisation × Gender × Homeschooling 1	−3.42 ***	0.96	−5.31–−1.53	<0.001
	Specialisation × Gender × Homeschooling 2	−5.58 ***	1.14	−7.81–−3.34	<0.001

*Notes.*^†^ *p* < 0.05; ** *p* < 0.01; *** *p* < 0.001. Gender coded −1 (Male), +1 = (Female). Homeschooling status 1 coded as 0 (voluntary homeschoolers), 1 (not homeschooling, i.e., child attending school in-person), and −1 (mandated homeschooling due to COVID-19). Homeschooling status 2 coded as −1 (voluntary homeschoolers), 0 (not homeschooling, i.e., child attending school in-person), and 1 (mandated homeschooling due to COVID-19).

**Table 10 ijerph-19-17021-t010:** Effects of equity and homeschooling status on male vs. female parents’ well-being in mixed-sex couples.

Outcome	Predictors	*B*	SE (*b*)	95% CI	*p*
**Depression** Marginal R^2^/Conditional R^2^ = 0.050/0.599 AIC/BIC = 9689.9/9792.1	Intercept	12.25 **	1.43	9.45–15.05	<0.001
	Parent age	−0.04	0.03	−0.11–0.02	0.185
	No. of children	−0.57 *	0.2	−0.97–−0.17	0.005
	Relationship length	−0.04	0.04	−0.11–0.04	0.316
	Family Income	−0.01	0.1	−0.21–0.19	0.902
	Education	−0.03	0.11	−0.26–0.19	0.767
	Equity	2.33	4.45	−6.39–11.05	0.601
	Gender	−0.67	0.48	−1.61–0.28	0.168
	Homeschooling 1	−0.48	0.25	−0.96–0.01	0.054
	Homeschooling 2	−0.74 *	0.29	−1.31–−0.18	0.009
	Gender × Equity	3.62	2.28	−0.85–8.09	0.112
	Gender × Homeschooling 1	0.24	0.12	−0.01–0.48	0.056
	Gender × Homeschooling 2	0.28 ^†^	0.14	0.01–0.55	0.040
	Equity × Homeschooling 1	−0.24	1.04	−2.28–1.81	0.822
	Equity × Homeschooling 2	−1.54	1.28	−4.04–0.96	0.228
	Equity × Gender × Homeschooling 1	−1.21 ^†^	0.54	−2.27–−0.15	0.025
	Equity × Gender × Homeschooling 2	−0.90	0.65	−2.18–0.39	0.170
**Anxiety** Marginal R^2^/Conditional R^2^ = 0.044/0.537 AIC/BIC = 9364.7/9467.0	Intercept	9.31 **	1.24	6.87–11.74	<0.001
	Parent age	−0.04	0.03	−0.09–0.02	0.195
	No. of children	−0.47 *	0.18	−0.82–−0.13	0.007
	Relationship length	−0.04	0.03	−0.10–0.02	0.207
	Family Income	0.00	0.09	−0.18–0.18	0.986
	Education	−0.09	0.1	−0.29–0.11	0.371
	Equity	4.53	3.85	−3.01–12.07	0.239
	Gender	−0.88	0.46	−1.78–0.01	0.052
	Homeschooling 1	−0.20	0.21	−0.62–0.22	0.346
	Homeschooling 2	−0.21	0.25	−0.70–0.28	0.398
	Gender × Equity	0.91	2.15	−3.31–5.13	0.674
	Gender × Homeschooling 1	0.31 *	0.12	0.08–0.54	0.008
	Gender × Homeschooling 2	0.34 *	0.13	0.08–0.60	0.009
	Equity × Homeschooling 1	−0.39	0.9	−2.16–1.37	0.663
	Equity × Homeschooling 2	−1.93	1.1	−4.09–0.24	0.081
	Equity × Gender × Homeschooling 1	−0.32	0.51	−1.32–0.68	0.527
	Equity × Gender × Homeschooling 2	−0.43	0.62	−1.65–0.78	0.483
**Perceived Stress** Marginal R^2^/Conditional R^2^ = 0.053/0.509 AIC/BIC = 7721.3/7823.5	Intercept	10.11 **	0.73	8.68–11.55	<0.001
	Parent age	−0.04 ^†^	0.02	−0.07–−0.01	0.014
	No. of children	−0.19	0.1	−0.39–0.01	0.069
	Relationship length	0.00	0.02	−0.04–0.04	0.935
	Family Income	−0.12 ^†^	0.05	−0.22–−0.01	0.028
	Education	−0.04	0.06	−0.17–0.08	0.475
	Equity	1.17	2.25	−3.25–5.59	0.603
	Gender	0.14	0.28	−0.41–0.68	0.625
	Homeschooling 1	−0.14	0.13	−0.39–0.11	0.265
	Homeschooling 2	−0.35 ^†^	0.15	−0.63–−0.06	0.018
	Gender × Equity	−0.31	1.32	−2.90–2.27	0.811
	Gender × Homeschooling 1	0.01	0.07	−0.13–0.15	0.847
	Gender × Homeschooling 2	0.06	0.08	−0.10–0.21	0.465
	Equity × Homeschooling 1	−0.46	0.53	−1.50–0.57	0.379
	Equity × Homeschooling 2	−0.52	0.65	−1.79–0.75	0.424
	Equity × Gender × Homeschooling 1	−0.15	0.31	−0.77–0.46	0.623
	Equity × Gender × Homeschooling 2	0.14	0.38	−0.60–0.89	0.707
**Relationship Satisfaction** Marginal R^2^/Conditional R^2^ = 0.064/0.790 AIC/BIC = 8622.4/8724.6	Intercept	10.96 **	1.2	8.60–13.32	<0.001
	Parent age	−0.03	0.03	−0.09–0.02	0.227
	No. of children	0.67 **	0.18	0.33–1.02	<0.001
	Relationship length	0.02	0.03	−0.04–0.09	0.444
	Family Income	−0.07	0.08	−0.23–0.09	0.389
	Education	−0.02	0.08	−0.17–0.13	0.820
	Equity	−8.32 ^†^	3.85	−15.86–−0.78	0.031
	Gender	0.36	0.29	−0.20–0.92	0.210
	Homeschooling 1	0.75 **	0.21	0.33–1.16	<0.001
	Homeschooling 2	0.65 *	0.25	0.17–1.13	0.009
	Gender × Equity	−0.55	1.35	−3.20–2.11	0.685
	Gender × Homeschooling 1	−0.14	0.07	−0.28–0.00	0.056
	Gender × Homeschooling 2	−0.13	0.08	−0.29–0.03	0.114
	Equity × Homeschooling 1	1.56	0.9	−0.21–3.33	0.084
	Equity × Homeschooling 2	3.15 *	1.1	0.99–5.32	0.004
	Equity × Gender × Homeschooling 1	0.29	0.32	−0.34–0.92	0.367
	Equity × Gender × Homeschooling 2	0.41	0.39	−0.35–1.17	0.288
**Relationship Conflict** Marginal R^2^/Conditional R^2^ = 0.071/0.670 AIC/BIC = 12,465.1/12,567.3	Intercept	38.01 **	3.55	31.06–44.97	<0.001
	Parent age	−0.08	0.08	−0.24–0.08	0.344
	No. of children	−1.40 *	0.51	−2.40–−0.40	0.006
	Relationship length	−0.15	0.09	−0.33–0.03	0.097
	Family Income	0.53 ^†^	0.25	0.03–1.03	0.037
	Education	−0.06	0.27	−0.58–0.47	0.836
	Equity	2.35	11.14	−19.48–24.18	0.833
	Gender	2.89 *	1.09	0.76–5.01	0.008
	Homeschooling 1	−2.48 **	0.62	−3.70–−1.27	<0.001
	Homeschooling 2	−3.58 **	0.72	−4.99–−2.18	<0.001
	Gender × Equity	−15.39 *	5.13	−25.45–−5.34	0.003
	Gender × Homeschooling 1	−0.37	0.28	−0.92–0.17	0.181
	Gender × Homeschooling 2	−0.79 ^†^	0.31	−1.40–−0.19	0.011
	Equity × Homeschooling 1	−1.40	2.61	−6.51–3.72	0.592
	Equity × Homeschooling 2	−2.15	3.2	−8.41–4.12	0.502
	Equity × Gender × Homeschooling 1	2.22	1.22	−0.16–4.60	0.068
	Equity × Gender × Homeschooling 2	4.56 *	1.47	1.67–7.45	0.002

*Notes.*^†^ *p* < 0.05; * *p* < 0.01; ** *p* < 0.001. Gender-coded −1 (Male), +1 = (Female). Homeschooling status 1 coded as 0 (voluntary homeschoolers), 1 (not homeschooling, i.e., child attending school in-person), and −1 (mandated homeschooling due to COVID-19). Homeschooling status 2 coded as −1 (voluntary homeschoolers), 0 (not homeschooling, i.e., child attending school in-person), and 1 (mandated homeschooling due to COVID-19).

## Data Availability

The data presented in this study are available in de-identified form on reasonable request from the corresponding author.

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
