# Peer review of "Division of Labour and Parental Mental Health and Relationship Well-Being during COVID-19 Pandemic-Mandated Homeschooling"

_ijerph, 2022, doi:10.3390/ijerph192417021_

Round 1

Reviewer 1 Report

Dear authors! Your work is very interesting and important. However, the conclusions and discussión should stress that statistical approach shows always only isolated part of reality. The aspects of well being should not be related only to economic reasons and income of teh families. People need independence, communication, broad social relations and responsabilites. The humankind at the present moment suffer a big crisis, which can't be explain and studies by statistical data. I suggest you only that the real life should be reduced to statistical data and that a lot of differential data must be taken into consideration for determination of what did people realy thought and did during pandemic. The topic of home schooling is especially difficult. What do childern know? Haw they sped thier time and how they communicate and how they make freinds during and after pandemic (after or still in)? You data ana analysis is important, but, please, let the reader know that this are not the only results and not the only way of explanation of reality.

Reviewer 2 Report

The work presents ample evidence that supports the approach of the research problem with updated sources and trying to be exhaustive in the review of the state of the art. They systematically present information, clearly state hypotheses and proposals for data analysis is appropriate, even when adjustments are required to different statistical analyses are justified and presented in a clear manner. 

The authors recognize the scope and limitations, to a large extent recognize that one of the main limitations of the work is the indirect measures of the distribution of work, the importance that when establishing an alternative avenue of research consider such objective measures to reduce error in behavioral estimates. 

This opens the possibility that many of the hypotheses accepted or rejected by the marginal, but significant relationships that have been obtained, could change the trend of the results and the possible conclusions that can be reached. 

Reviewer 3 Report

First of all, I would like to thank you for the opportunity to review the article submitted to the International Journal of Environmental Research and Public Health.
You have written an interesting study, which is very timely and focuses on a really important issue. I have no doubt that this article deserves to be published.

Congratulations of your hard work.
I only have a few editorial notes:

1. It is recommended not to repeat in keywords the words already included in the title, for a better localization of the scope.
2. It is proposed to use the word "gender" instead of "sex" because gender is a much more ambiguous concept today.
3. There are a few minor errors in the text, e.g. the word "and" is in italics (line 108, line 147, line 390).
4. The statistical significance level, i.e.: “p” symbol should be in italics.
5. The “Introduction” section is too long. You should shorten it.

Best regards.
